# Multi Timescale Stochastic Approximation: Stability and Convergence

## Abstract

This paper presents the first sufficient conditions that guarantee the stability and almost sure convergence of multi-timescale stochastic approximation (SA) iterates. It extends the existing results on one-timescale and two-timescale SA iterates to general $N$-timescale stochastic recursions, for any $N \geq 1$, using the ordinary differential equation (ODE) method. As an application of our results, we first study stochastic approximation algorithms augmented with heavy-ball momentum in the context of Gradient Temporal Difference (GTD) learning. The addition of momentum introduces an auxiliary state that evolves on an intermediate timescale, resulting in a three-timescale recursion. We show that when the momentum parameters are chosen appropriately, the resulting scheme fits within our framework and converges almost surely to the same fixed point as the baseline GTD algorithm. The stability and convergence of all iterates, including the momentum state, are guaranteed by our main results, without requiring ad hoc bounds. We then study off-policy actor–critic algorithms with a baseline learner, actor, and critic updated on separate timescales. In contrast to prior work, we eliminate projection steps from the actor update and instead use our convergence framework to guarantee stability and almost sure convergence of all components. Finally, we extend our analysis to constrained policy optimization in the average-reward setting, where the actor, critic, and dual variables evolve on three distinct timescales, and we verify that the resulting dynamics satisfy the conditions of our general theorem. Together, these examples demonstrate how diverse reinforcement learning algorithms spanning momentum acceleration, off-policy learning, and primal–dual methods fit naturally into the proposed multi-timescale framework.

## 1 Introduction

Stochastic Approximation (SA) algorithms (Robbins and Monro, 1951) are a class of iterative procedures used to find the zeros of a function $h : \mathbb{R}^d \to \mathbb{R}^d$, when the actual function is not known but noisy observations of the same are available. Standard stochastic approximation iterates are as given below:

$$x_{n+1} = x_n + \alpha_n f(x_n, Y_n), n \geq 0, \tag{1}$$

where, $x_n \in \mathbb{R}^d$, $n \geq 0$, is a sequence of parameters updated as in (1) using $f(x_n, Y_n)$ that are noisy observations of the true function value $h(x_n)$. Further, $Y_n$, $n \geq 0$, is the corresponding noise sequence and $\alpha_n, n \geq 0$, the given step-size sequence. In this paper, we assume that the noisy observation of the function is of the following form:

$$f(x_n, Y_n) = h(x_n) + M_{n+1}, \tag{2}$$

where $h(\cdot)$ is the objective function and $M_{n+1}, n \geq 0$, is a martingale difference sequence with respect to the increasing sequence of sigma fields $\mathcal{F}_n = \sigma(x_i, M_i, 0 \leq i \leq n)$, $n \geq 0$. The form (2) of $f$ is fairly general. For instance, if $Y_n \neq 0$, are i.i.d., one may let $h(x_n) = E_Y[f(x_n, Y)]$ and $M_{n+1} = f(x_n, Y_n) - h(x_n)$, $n \geq 0$, are the corresponding martingale differences. Then (2) can be seen to hold.

In many applications, the objective function $h$ might require an additional averaging for it's evaluation at a given parameter update. The procedure, in such a case, would require an additional recursion to estimate $h$

and the overall procedure would result in a recursive computation in nested loops. For example, the policy iteration procedure in Markov decision processes (Puterman, 2014), involves two nested loops in which the outer loop performs policy improvement while the inner loop performs policy evaluation for a given policy update. Actor-critic algorithms (Konda and Tsitsiklis, 2000; Bhatnagar et al., 2009) are stochastic recursive iterates that try to mimic the nested loop structure of policy iteration under noisy sample estimates where the actor recursion performs policy improvement while the critic is responsible for policy evaluation. This is achieved by making use of two different step-size schedules, one of which goes to zero at a rate faster than the other. The recursion governed by the step-size that goes to zero faster ends up being the 'slower' timescale recursion while the one governed by the other step-size schedule corresponds to the 'faster' timescale update. Thus, even though both recursions are executed simultaneously, at each instant, one sees a similar effect that a nested loop procedure would provide. In particular, the faster recursion sees the slower update as 'quasi-static' while the latter sees the faster recursion as essentially 'equilibrated'.

Another example along these lines would be of a Markov reward process (Marbach and Tsitsiklis, 2001) whose transition probabilities depend on a certain parameter $\theta$ and the goal is to find the optimal parameter, i.e., the one that maximizes the long-run average reward. Two-timescale stochastic iterates can be used to solve such problems as well (Bhatnagar et al., 2013) where the recursion corresponding to the inner loop, i.e., the one governed by the faster timescale, performs the averaging of the single-stage rewards for a given update of the parameter $\theta$ while the outer loop recursion or the one governed by the slower timescale performs a maximization of the average rewards over $\theta$. Two-timescale stochastic approximation algorithms have the following general form:

$$x_{n+1}^{(1)} = x_n^{(1)} + \alpha_n^{(1)} \left( h^{(1)} \left( x_n^{(1)}, x_n^{(2)} \right) + M_{n+1}^{(1)} \right), \tag{3}$$

$$x_{n+1}^{(2)} = x_n^{(2)} + \alpha_n^{(2)} \left( h^{(2)} \left( x_n^{(1)}, x_n^{(2)} \right) + M_{n+1}^{(2)} \right), \tag{4}$$

where $\{M_{n+1}^{(1)}\}$ and $\{M_{n+1}^{(2)}\}$ are both martingale difference sequences and $\{\alpha_n^{(1)}\}$ and $\{\alpha_n^{(2)}\}$ are the step-size sequences, respectively. A specific requirement of the step-sizes for (3)-(4) to be a two-timescale recursion is that $\frac{\alpha_n^{(1)}}{\alpha_n^{(2)}} \to 0$ as $n \to \infty$.

These algorithms have been analyzed for their convergence in Borkar (1997) assuming, in particular, that the iterates remain stable, i.e., that almost surely $\sup_n \| x_n^{(1)} \| < \infty$ and $\sup_n \| x_n^{(2)} \| < \infty$, respectively. This is a nontrivial requirement and in fact until recently, there were no known conditions to verify this requirement. In the case of one-timescale algorithms, i.e., when $N = 1$, sufficient conditions for stability and convergence of stochastic iterates are available in the literature. For instance, in (Borkar and Meyn, 2000; Borkar, 2008), certain conditions are provided on the limiting ODE and its scaling limit that are shown to be sufficient for the stability of the recursion.

In Lakshminarayanan and Bhatnagar (2017), the stability conditions provided in Borkar and Meyn (2000) are generalized to the two-timescale setting and these happen to be the first set of conditions that guarantee stability of general two-timescale stochastic recursions and along with Borkar (1997), provide a complete set of conditions under which the recursions given by equation 3 and equation 4 converge almost surely as $n \to \infty$. In this paper, we generalize the results of Lakshminarayanan and Bhatnagar (2017) to the case of $N$-timescale stochastic approximation for any general $N \geq 1$. We explain below the motivation for doing so.

In many problems of optimization or control under uncertainty, one encounters algorithms involving three or more timescale recursions. Such examples arise for instance when designing actor-critic algorithms for constrained Markov decision processes (Altman, 1999), see for instance, Borkar (2005); Bhatnagar (2010) or hierarchical reinforcement learning (Bhatnagar and Panigrahi, 2006). In Borkar (2005); Bhatnagar (2010), the constraints are relaxed using a Lagrangian and the Lagrange parameter is updated on the slowest timescale, while the actor (policy improvement) update happens on the middle timescale and the critic (policy evaluation) update happens on the fastest timescale. Further, for settings such as hierarchical reinforcement learning (Bhatnagar and Panigrahi, 2006), one may encounter algorithms with a number of timescales that is proportionate to the number of levels of decision making. Sufficient conditions for stability and convergence of algorithms (with $N > 2$) are not available in the literature. The results we present here are general enough and applicable to such settings. In fact, we also study, in this paper, applications involving certain

reinforcement learning algorithms with momentum that require three and four timescale recursions whose convergence we prove by verifying that our sufficient conditions hold.

We provide the first set of sufficient conditions that ensure both the stability and convergence of general $N$-timescale stochastic approximation recursions. Our conditions are obtained by generalizing the requirements in Borkar and Meyn (2000) for one-timescale algorithms as well as in Lakshminarayanan and Bhatnagar (2017) for two-timescale recursions. We provide the full analysis of stability and convergence of a general $N$-timescale stochastic recursion using the aforementioned sufficient conditions. We then demonstrate the usefulness of these conditions in the context of three and four timescale gradient temporal difference learning algorithms in reinforcement learning. Such an analysis is made possible because of the sufficient conditions that we provide for the stability and convergence of $N$-timescale algorithms for any $N \geq 1$. Since $N \geq 1$ is arbitrary, we believe that the sufficient conditions for stability and convergence of $N$-timescale stochastic recursions will be extremely useful for algorithm designers who will henceforth be able to prove the stability and convergence of the stochastic recursions through verification of the aforementioned sufficient conditions.

The rest of the paper is organized as follows: In Section 2, we present the notation used in the paper as well as our assumptions. Here, for improved clarity, we first present our assumptions for the 3-timescale case before we present the general assumptions for $N$-timescale stochastic iterates. In Section 3, we present our main results on stability and convergence of $N$-timescale SA. We then provide brief descriptions of several applications of our results in reinforcement learning. Next in Section 4, we provide a proof sketch of the main results with a complete analysis of stability and convergence of the recursions in Appendix A. In Section 5, we present complete stability and convergence analysis of several reinforcement learning algorithms using our new results for multi-timescale updates. We first start with a policy evaluation algorithm, Gradient Temporal Difference (GTD) learning with heavy-ball momentum, where the momentum term introduces an intermediate timescale and yields a three-timescale recursion. When the momentum parameters are properly chosen, our framework guarantees almost sure convergence of all iterates—including the momentum state—to the standard GTD solution without requiring additional boundedness assumptions. We also show empirically that the proposed momentum methods outperform their vanilla counterparts. Next we apply our framework to off-policy actor–critic algorithms with a baseline, where the actor, critic, and baseline updates evolve on separate timescales. Without using projections, we show that the iterates remain stable and converge under standard assumptions by verifying that the recursion satisfies our structural conditions. Finally, we extend our analysis to constrained policy optimization in the average-reward setting, where the actor, critic, and dual variables evolve on three distinct timescales, and we verify that the resulting dynamics satisfy the conditions of our general theorem. We present concluding remarks as well as potential directions for future research in Section 6.

## 2  Notations and Assumptions

In this section, we carefully setup the notation followed in the entire paper and then explicitly provide the assumptions under which the iterates are stable and convergent. The $N$ stochastic recursions are given by:

$$
\begin{aligned}
x_{n+1}^{(1)} &= x_n^{(1)} + \alpha_n^{(1)}(h^{(1)}(x_n^{(1)}, x_n^{(2)}, \ldots, x_n^{(N)}) + M_{n+1}^{(1)}), \\
x_{n+1}^{(2)} &= x_n^{(2)} + \alpha_n^{(2)}(h^{(2)}(x_n^{(1)}, x_n^{(2)}, \ldots, x_n^{(N)}) + M_{n+1}^{(2)}), \\
&\vdots \\
x_{n+1}^{(N)} &= x_n^{(N)} + \alpha_n^{(N)}(h^{(N)}(x_n^{(1)}, x_n^{(2)}, \ldots, x_n^{(N)}) + M_{n+1}^{(N)}).
\end{aligned}
\tag{5}
$$

Here, $\{x_n^{(j)} \in \mathbb{R}^{d_j}\}_{1 \leq j \leq N}$ are the $N$ parameters that are updated at each time-step and the subscript $n$ denotes the time-step or iteration-index of the update. $\{\alpha_n^{(j)}, n \geq 0\}$, $1 \leq j \leq N$ are $N$ different step-size sequences. The functions $h^{(j)} : \mathbb{R}^{d_1+d_2+\cdots+d_N} \to \mathbb{R}^{d_j}$, $1 \leq j \leq N$ could potentially depend on all the $N$ parameters. Also, $\{M_{n+1}^{(j)}\}$, $1 \leq j \leq N$ are the associated martingale difference noise sequences.

The following are standard assumptions made while analyzing stochastic recursions:

**(A:1)** $h^{(j)}$, $1 \leq j \leq N$ are Lipschitz continuous functions.

**(A:2)** $\{M_{n+1}^{(j)}\}$, $1 \leq j \leq N$ are $N$ martingale difference sequences with respect to the increasing sequence of sigma fields $\{\mathcal{F}_n\}$, where $\mathcal{F}_n = \sigma\left(x_i^{(j)}, M_{i+1}^{(j)}, 1 \leq j \leq N, i \leq n\right)$, $n \geq 0$, with $\mathbb{E}[\|M_{n+1}^{(i)}\|^2 | \mathcal{F}_n] \leq K^{(i)}(1 + \sum_{j=1}^{N} \|x_n^{(j)}\|^2), n \geq 0$, for some constant $K^{(i)} > 0, 1 \leq i \leq N$.

**(A:3)** $\{\alpha_n^{(j)}\}_{1 \leq j \leq N}$ are $N$ step-size schedules that satisfy the following:

(i) $\alpha_n^{(j)} > 0, \forall n, \sum_n \alpha_n^{(j)} = \infty, 1 \leq j \leq N$,

(ii) $\sum_{n=0}^{\infty} \sum_{j=1}^{N} (\alpha_n^{(j)})^2 < \infty$,

(iii) $\frac{\alpha_n^{(j)}}{\alpha_n^{(j-1)}} \to 0$, as $n \to \infty, 1 < j \leq N$.

Assumptions **(A:1)**-**(A:3)** along with stability of the iterates, i.e., $\sup_n(x_n^{(j)}) < \infty$, $\forall 1 \leq j \leq N$ can be shown to ensure convergence. The next set of assumptions along with **(A:1)**-**(A:3)** will provide sufficient conditions for stability of the iterates. For the case of one-timescale stochastic update recursions, Borkar and Meyn (2000) provided verifiable sufficient conditions for stability of the iterate sequence (see (A1) in Borkar and Meyn (2000) or (A5) in Chapter 3 of Borkar (2008)). By considering a scaled version of the original recursion and a scaled ODE associated with the scaled version, Borkar and Meyn (2000) shows that under sufficiently general conditions, the original recursion remains stable.

For two-timescale recursions, Lakshminarayanan and Bhatnagar (2017) provided conditions for stability (see (A4) and (A5) there). Since the iterates, in this case, evolve along both the timescales simultaneously, Lakshminarayanan and Bhatnagar (2017) analyzed the rescaled trajectories of both the iterates and presented the aforementioned conditions (cf. (A4) and (A5) therein), one for each timescale, to ensure stability. We extend these ideas to the general $N$-timescale regime, where we analyze the $N$ rescaled iterates and come up with a set of $N$ conditions on the behaviour of the limiting ODEs corresponding to the rescaled recursions. For improved clarity we first state these conditions in the 3-Timescale setting in Sections 2.1. In Section 5, we analyze the stability and convergence of a reinforcement learning algorithms with momentum that involves three timescale updates by verifying these conditions in Sections 2.1. We state the conditions for general $N$-timescale stochastic recursions in Section 2.2. The assumptions in what follows will be numbered as **(B.j.i)** and **(C.j.i)** respectively. Here the **B**-assumptions concern the rescaled ODEs while the **C**-assumptions are for the ODEs corresponding to the original recursions. Further, the index **j** refers to the number of timescales used in the algorithm and **i** takes values in general between **1** and **j**.

### 2.1  The 3-Timescale Case

We first state the conditions that ensure stability and convergence for the $N = 3$ case. In Section 5 we discuss an application of our results to the case of stochastic approximation with momentum where the iterates can be analyzed using the results for the 3-Timescale case.

**(B.3.1)** For $c \geq 1$, define the following scaled functions based on $h^{(1)}$:

$$h_c^{(1)}(x^{(1)}, x^{(2)}, x^{(3)}) = \frac{h^{(1)}(cx^{(1)}, cx^{(2)}, cx^{(3)})}{c}.$$

Further, $h_c^{(1)} \to h_\infty^{(1)}$ as $c \to \infty$ uniformly on compacts. The ODE $\dot{x}^{(1)}(t) = h_\infty^{(1)}(x^{(1)}(t), x^{(2)}, x^{(3)})$, for $x^{(j)} \in \mathbb{R}^{d_j}$, $j = 2, 3$, has a unique globally asymptotically stable equilibrium $\lambda_\infty^{(1)}(x^{(2)}, x^{(3)})$, where $\lambda_\infty^{(1)} : \mathbb{R}^{d_2+d_3} \to \mathbb{R}^{d_1}$ is Lipschitz continuous. Further $\lambda_\infty^{(1)}(0, 0) = 0$.

**(B.3.2)** For $c \geq 1$, define the following scaled functions based on $h^{(2)}$:

$$h_c^{(2)}(x^{(2)}, x^{(3)}) = \frac{h^{(2)}\left(c\lambda_\infty^{(1)}(x^{(2)}, x^{(3)}), cx^{(2)}, cx^{(3)}\right)}{c}.$$

Further, $h_c^{(2)} \to h_\infty^{(2)}$ as $c \to \infty$ uniformly on compacts. The ODE $\dot{x}^{(2)}(t) = h_\infty^{(2)}(x^{(2)}(t), x^{(3)})$, for $x^{(j)} \in \mathbb{R}^{d_j}$, $j = 3, 4$, has a unique globally asymptotically stable equilibrium $\lambda_\infty^{(2)}(x^{(3)})$, where $\lambda_\infty^{(2)} : \mathbb{R}^{d_3} \to \mathbb{R}^{d_2}$ is Lipschitz continuous. Further $\lambda_\infty^{(2)}(0) = 0$.

**(B.3.3)** For $c \geq 1$, define the following scaled functions based on $h^{(3)}$:

$$h_c^{(3)}\big(x^{(3)}\big) = \frac{h^{(3)}\left(c\lambda_\infty^{(1)}\big(\lambda_\infty^{(2)}(x^{(3)}), x^{(3)}\big), c\lambda_\infty^{(2)}(x^{(3)}), cx^{(3)}\right)}{c}.$$

Further, $h_c^{(3)} \to h_\infty^{(3)}$ as $c \to \infty$ uniformly on compacts. The ODE $\dot{x}^{(3)}(t) = h_\infty^{(3)}(x^{(3)}(t))$ has the origin in $\mathbb{R}^{d_3}$ as its unique globally asymptotically stable equilibrium.

**(C.3.1)** The ODE

$$\dot{x}^{(1)}(t) = h^{(1)}\big(x^{(1)}(t), x^{(2)}, x^{(3)}\big),$$

$x^{(j)} \in \mathbb{R}^{d_j}$, $j = 2, 3$, has a unique globally asymptotically stable equilibrium $\lambda^{(1)}(x^{(2)}, x^{(3)})$ where $\lambda^{(1)} : \mathbb{R}^{d_2 + d_3} \to \mathbb{R}^{d_1}$ is Lipschitz continuous.

**(C.3.2)** The ODE

$$\dot{x}^{(2)}(t) = h^{(2)}\Big(\lambda^{(1)}\big(x^{(2)}(t), x^{(3)}\big), x^{(2)}(t), x^{(3)}\Big),$$

with $x^{(3)} \in \mathbb{R}^{d_3}$, has a unique globally asymptotically stable equilibrium $\lambda^{(2)}(x^{(3)})$ and $\lambda^{(2)} : \mathbb{R}^{d_3} \to \mathbb{R}^{d_2}$ is Lipschitz continuous.

**(C.3.3)** The ODE

$$\dot{x}^{(3)}(t) = h^{(3)}\Big(\lambda^{(1)}\big(\lambda^{(2)}(x^{(3)}(t)), x^{(3)}(t)\big), \lambda^{(2)}(x^{(3)}(t)), x^{(3)}(t)\Big),$$

has a unique globally asymptotically stable equilibrium $x_*^{(3)} \in \mathbb{R}^{d_3}$.

## 2.2 The N-Timescale Case

Next we generalize these assumptions to the $N$-timescale regime. The assumptions on the timescales $1, 2, \ldots, N$ are compactly stated as assumptions **(B.N.i)** and **(C.N.i)**, respectively, where in fact the index $i$ takes $N$ values, i.e., $i = 1, \ldots, N$. Thus, these conditions encapsulate a total of $2N$ assumptions, with $N$ B-assumptions and $N$ C-assumptions, respectively. In addition, we shall also require Assumptions A:1-A:3 stated previously.

**(B.N.i)** For $c \geq 1$, define the following scaled functions based on $h^{(i)}$:

$$h_c^{(i)}\left(x^{(i)}, x^{(i+1)}, \ldots, x^{(N)}\right) = \frac{h^{(i)}(cy^{(1)}, cy^{(2)}, \ldots, cy^{(N)})}{c},$$

where, for $j < i$, $\quad y^{(j)} = \lambda_\infty^j\left(\lambda_\infty^{j+1}\left(\ldots\lambda_\infty^{(i-3)}\big(\lambda_\infty^{(i-2)}\big(\lambda_\infty^{(i-1)}(x^{(i)}, \ldots, x^{(N)}), x^{(i)}, \ldots, x^{(N)}\big),\right.\right.$

$$\left.\left.\lambda_\infty^{(i-1)}(x^{(i)}, \ldots, x^{(N)}), x^{(i)}, \ldots, x^{(N)}\big), \ldots, x^{(i)}, \ldots, x^{(N)}\right)\right)$$

and for $j \geq i$ $y^{(j)} = cx^{(j)}$. Further, $h_c^{(i)} \to h_\infty^{(i)}$ as $c \to \infty$ uniformly on compacts. The ODE

$$\dot{x}^{(i)}(t) = h_\infty^{(i)}(x^{(i)}(t), x^{(i+1)}, \ldots, x^{(N)}), \tag{6}$$

for $x^{(j)} \in \mathbb{R}^{d_j}, j \in \{i+1, \ldots, N\}$ has

(i) a unique globally asymptotically stable equilibrium $\lambda_\infty^{(i)}(x^{(i+1)}, x^{(i+2)}, \ldots, x^{(N)})$, $i = 1, \ldots, N-1$, where each $\lambda_\infty^{(i)} : \mathbb{R}^{d_{i+1} + d_{i+2} + \cdots + d_N} \to \mathbb{R}^{d_i}$ is Lipschitz continuous. Further, $\lambda_\infty^{(i)}(0, 0, \ldots, 0) = 0, \forall i = 1, \ldots, N-1$, and

(ii) for $i = N$, the origin in $\mathbb{R}^{d_N}$ is the unique globally asymptotically stable equilibrium of (6).

Chapter 6 of Borkar (2008) makes two more assumptions namely (A1) and (A2) there regarding the global asymptotic stable equilibria of the ODE trajectories along the two timescales. We make $N$ such assumptions here (to account for the $N$ different timescales in our recursions) on the ODE trajectories and compactly write these as **(C.N.i)**, $1 \leq i \leq N$.

**(C.N.i)** The ODE $\dot{x}^{(i)}(t) = h^{(i)}\big(z^{(1)}(t), z^{(2)}(t), \ldots, z^{(N)}(t)\big)$, has a globally asymptotically stable equilibrium:
(i)$\lambda(x^{i+1}, \ldots, x^N)$, $x^{(j)} \in \mathbb{R}^{d_j}, i+1 \leq j \leq N$ with $1 \leq i \leq N-1$, and (ii) $x_*^{(N)} \in \mathbb{R}^{d_N}$.

Here, for $j \geq i$, $z^{(j)}(t) = x^{(j)}.$, and for $j < i$,
$$z^{(j)}(t) = \lambda^j\bigg(\lambda^{j+1}\Big(\ldots\lambda^{(i-3)}\big(\lambda^{(i-2)}\big(\lambda^{(i-1)}(x^{(i)}(t), x^{(i+1)}\ldots, x^{(N)}), x^{(i)}(t), x^{(i+1)}, \ldots, x^{(N)}\big),$$
$$\lambda^{(i-1)}(x^{(i)}(t), x^{(i+1)}, \ldots, x^{(N)}), x^{(i)}(t), x^{(i+1)}, \ldots, x^{(N)}\big), \ldots, x^{(i)}(t), x^{(i+1)}, \ldots, x^{(N)}\Big)\bigg)$$

## 3  Main Results

We state in this section (and later prove) our results for general $N$-timescale stochastic approximation algorithms as given in Section 2.2.

**Theorem 1.** *We define for* $1 \leq j \leq N-2$, $\lambda^{(j:N-1)}(x) \stackrel{\triangle}{=} \lambda^{(j)}\Big(\lambda^{(j+1)}\big(\ldots\lambda^{(N-2)}(\lambda^{(N-1)}(x), x), \ldots, x\big), x\Big)$.
*Under the assumptions (A:1)-(A:3), (B.N.i)$_{1 \leq i \leq N}$ and (C.N.i)$_{1 \leq i \leq N}$,*
$$x_n^{(1)} \to x_*^{(1)} = \lambda^{(1:N-1)}(x_*^{(N)}),$$
$$x_n^{(2)} \to x_*^{(2)} = \lambda^{(2:N-1)}(x_*^{(N)}),$$
$$\vdots$$
$$x_n^{(N-1)} \to x_*^{(N-1)} = \lambda^{(N-1:N-1)}(x_*^{(N)}),$$
$$x_n^{(N)} \to x_*^{(N)}.$$

We first prove that the $N$ iterates are convergent under the assumption that the $N$ iterates are stable. Specifically, we first prove a similar result (see Theorem 2) under the following assumption instead of **(B.N.i)**, $1 \leq i \leq N$.

**(B.N.N+1)** The following holds for the iterates: $\sup_n \left( \|x_n^{(1)}\| + \|x_n^{(2)}\| + \cdots + \|x_n^{(N)}\| \right) < \infty$   a.s.

**Theorem 2.** *Under the assumptions (A:1)-(A:3), (B.N.N+1) and (C.N.i)$_{1 \leq i \leq N}$,*
$$x_n^{(1)} \to x_*^{(1)} = \lambda^{(1:N-1)}(x_*^{(N)}), \quad x_n^{(2)} \to x_*^{(2)} = \lambda^{(2:N-1)}(x_*^{(N)}),$$
$$\vdots$$
$$x_n^{(N-1)} \to x_*^{(N-1)} = \lambda^{(N-1:N-1)}(x_*^{(N)}), \quad x_n^{(N)} \to x_*^{(N)}.$$

Next, we shall show that the $N$ recursions are a.s. stable under the assumptions **(B.N.i)**$_{1 \leq i \leq N}$. In other words, Assumption **(B.N.N+1)** holds under **(B.N.i)**$_{1 \leq i \leq N}$.

**Theorem 3.** *Under the assumptions (A:1)-(A:3), (B.N.i)$_{1 \leq i \leq N}$ and (C.N.i)$_{1 \leq i \leq N}$,*
$$\sup_n \left( \|x_n^{(1)}\| + \|x_n^{(2)}\| + \cdots + \|x_n^{(N)}\| \right) < \infty.$$

Theorem 3 with Theorem 2 imply that the recursions converge a.s. under the assumptions in Theorem 1. Finally, we consider the $N$-timescale recursions where each iterate contains a small additive perturbation term as follows:

$$x_{n+1}^{(1)} = x_n^{(1)} + \alpha_n^{(1)}(h^{(1)}(x_n^{(1)}, x_n^{(2)}, \ldots, x_n^{(N)}) + M_{n+1}^{(1)} + \varepsilon_n^{(1)}),$$
$$x_{n+1}^{(2)} = x_n^{(2)} + \alpha_n^{(2)}(h^{(2)}(x_n^{(1)}, x_n^{(2)}, \ldots, x_n^{(N)}) + M_{n+1}^{(2)} + \varepsilon_n^{(1)})),$$
$$\vdots$$
$$x_{n+1}^{(N)} = x_n^{(N)} + \alpha_n^{(N)}(h^{(N)}(x_n^{(1)}, x_n^{(2)}, \ldots, x_n^{(N)}) + M_{n+1}^{(N)} + \varepsilon_n^{(1)})).$$

**Theorem 4.** *Assume **(A:1)-(A:3)**, **(B.N.i)**$_{1 \leq i \leq N}$ and **(C.N.i)**$_{1 \leq i \leq N}$ hold. Further assume that $\sum_{i=1}^{N} \varepsilon_n^{(i)} \to 0$ as $n \to \infty$. Then $x_n^{(j)}, 1 \leq j \leq N$ converges almost surely to the same solution as in Theorem 1.*

## 3.1 Applications of the General Theory

We now show how the abstract $N$-timescale convergence theorems yield concrete guarantees for three widely used reinforcement learning algorithms. In each case, we: (1) describe the objective being solved, (2) express the algorithm as a coupled stochastic recursion, and (3) map it to our general theory to obtain convergence results. A detailed convergence analysis of the algorithms can be found in Section 5.

1. **GTD2 with momentum (policy evaluation).** The goal is to evaluate a fixed policy $\pi$ using off-policy data collected from a behaviour policy $\mu$. Let $v^\pi$ denote the value function of $\pi$. GTD2 seeks to find the best linear approximation $v_\theta(x) = \phi(x)^\top \theta$ to $v^\pi$, solving the projected Bellman equation while correcting the instability introduced by off-policy sampling. A heavy-ball momentum term is added to the recursion to improve empirical convergence rate.

   **Stochastic recursion.** The algorithm maintains three iterates, $v_t$, a fast auxiliary vector used for temporal-difference errors, $u_t$ a momentum term tracking smoothed value estimates, $\theta_t$, the main parameter approximating the value function. The updates evolve on three distinct timescales $(\xi_t, \beta_t, \varrho_t)$ satisfying $\xi_t > \beta_t > \varrho_t$, and follow linear recursions:

   $$v_{t+1} = v_t + \xi_t(h^{(1)}(v_t, u_t, \theta_t) + M_{t+1}^{(1)}),$$
   $$u_{t+1} = u_t + \beta_t(h^{(2)}(v_t, u_t, \theta_t) + M_{t+1}^{(2)}),$$
   $$\theta_{t+1} = \theta_t + \varrho_t(h^{(3)}(v_t, u_t, \theta_t) + M_{t+1}^{(3)}).$$

   The functions $h^{(i)}$ are affine in their arguments and the noise terms $M_{t+1}^{(i)}$ are martingale differences satisfying standard boundedness and moment conditions.

   **Application of theory.** Each $h^{(i)}$ is globally Lipschitz, and the joint drift satisfies the hierarchical contraction properties required by Assumptions **(B)** and **(C)**. The stepsize conditions match Assumption **(A.2)**, and the feature/reward boundedness ensures that Assumption **(A.1)** holds. Consequently, our Theorem 1 implies:

   $$\theta_t \to \theta^* = -\bar{A}^{-1}\bar{b} \quad \text{almost surely,}$$

   where $(\bar{A}, \bar{b})$ are the expected GTD2 update matrices under the stationary distribution of $\mu$. The fast and intermediate iterates $(v_t, u_t)$ track their nested fixed points, with no additional stability constraints needed.

2. **Off-policy actor–critic (unconstrained policy optimization).** The objective is to maximise the expected discounted return $J(\pi)$ over a class of parameterised stochastic policies $\pi_\theta$, using samples drawn from an off-policy distribution. To stabilise training, the method uses a critic to estimate action-values (or advantages) and updates the actor via gradient ascent. An additional

auxiliary parameter may be used to track target quantities (e.g., baselines or bootstrapped TD targets), yielding a three-timescale algorithm.

**Stochastic recursion.** The algorithm maintains three parameters, a fast critic state $v_t$ (e.g., value or TD-error vector), an intermediate auxiliary estimate $u_t$ (e.g., for baselines or GTD stability), a slow actor parameter $\theta_t$ optimizing $J(\theta)$ via gradient feedback. The updates evolve on three nested timescales $\xi_t > \beta_t > \alpha_t$, and follow the recursion:

$$v_{t+1} = v_t + \xi_t\big(h^{(1)}(v_t, u_t, \theta_t) + M_{t+1}^{(1)}\big),$$
$$u_{t+1} = u_t + \beta_t\big(h^{(2)}(v_t, u_t, \theta_t) + M_{t+1}^{(2)}\big),$$
$$\theta_{t+1} = \theta_t + \alpha_t\big(h^{(3)}(v_t, u_t, \theta_t) + M_{t+1}^{(3)}\big).$$

The critic $v_t$ tracks a TD-style value estimate; $u_t$ may track an advantage baseline or auxiliary statistics; $\theta_t$ implements a policy gradient step using critic feedback.

**Application of theory.** All updates are Lipschitz and structured to obey a hierarchical contraction condition, so assumptions **(B)** and **(C)** hold. Boundedness of features and noise ensures assumptions **(A.1)** and **(A.3)**. The step-size hierarchy satisfies **(A.2)**. Then by Theorem 1:

$$v_t \to \lambda^{(1)}(u_t, \theta_t), u_t \to \lambda^{(2)}(\theta_t), \theta_t \to \theta^* \text{ where } \nabla_\theta J(\theta^*) = 0, \quad \text{almost surely.}$$

Thus the actor–critic scheme converges to a stationary point of the return objective, even under off-policy sampling, with all internal auxiliary states tracking their fixed points.

3. **Constrained actor–critic (policy optimization with constraints).** The goal is to solve a constrained reinforcement learning problem of the form

$$\max_\pi J(\pi) \quad \text{subject to } C_i(\pi) \leq \nu_i, \quad 1 \leq i \leq m,$$

where both the objective and constraints depend on expectations over trajectories. The method uses Lagrangian dual ascent: an inner loop updates the actor and critic, and an outer loop adjusts the Lagrange multipliers $\gamma_t$.

**Stochastic recursion.** The algorithm maintains a critic $(v_t)$ on the fastest timescale for value estimation, an actor $\theta_t$ on a slower timescale for primal ascent and dual variables $\gamma_t$ on the slowest timescale for constraint enforcement. These obey a three-timescale recursion of the form:

$$v_{t+1} = v_t + \beta_t\big(h^{(1)}(v_t, \theta_t, \gamma_t) + M_{t+1}^{(1)}\big),$$
$$\theta_{t+1} = \theta_t + \alpha_t\big(h^{(2)}(v_t, \theta_t, \gamma_t) + M_{t+1}^{(2)}\big),$$
$$\gamma_{t+1} = \gamma_t + \eta_t\big(h^{(3)}(\theta_t, \gamma_t) + M_{t+1}^{(3)}\big),$$

where $\beta_t > \alpha_t > \eta_t$ and the updates ascend the Lagrangian $\mathcal{L}(\theta, \gamma) = J(\theta) - \sum_i \gamma_i(C_i(\theta) - \nu_i)$.

**Application of theory.** Under standard assumptions (bounded features, Lipschitz dynamics, monotone constraint violation), the triple recursion satisfies all the conditions of Theorem 1. Then, the critic tracks its fixed point for each $(\theta_t, \gamma_t)$, the actor follows a projected gradient flow for $\mathcal{L}(\cdot, \gamma_t)$, the dual variables ascend the dual function to enforce constraints. The iterates converge to a saddle point of the Lagrangian, yielding a primal-dual pair $(\theta^*, \gamma^*)$ such that:

$$\nabla_\theta \mathcal{L}(\theta^*, \gamma^*) = 0, \qquad \gamma_i^* \geq 0, \quad C_i(\theta^*) \leq \nu_i, \quad \gamma_i^*(C_i(\theta^*) - \nu_i) = 0.$$

This satisfies the KKT conditions for the constrained problem.

## 4   Proof Sketch of Main Results

We provide a proof sketch of our main result, Theorem 1 here with a detailed proof provided in Appendix A.

1. **Reduction to a stability–convergence decomposition.** The argument splits naturally into two logically independent parts. First, assuming that the entire iterate vector $(x_n^{(1)}, x_n^{(2)}, \ldots, x_n^{(N)})$ remains almost surely bounded, we demonstrate that the $N$ coupled stochastic-approximation (SA) recursions converge to the prescribed hierarchical fixed point described in the theorem statement. Second, we remove that boundedness assumption by proving the required *a.s.* stability of all coordinates directly from the drift conditions in Assumptions **(B.N.i)**$_{1 \leq i \leq N}$. The two ingredients, taken together, yield the full claim.

2. **ODE limit for the fastest scale under boundedness.** Fix the fastest stepsize sequence $\{\alpha_n^{(1)}\}$. After dividing the $j$th recursion by the common factor $\alpha_n^{(1)}$ we observe that, for every $2 \leq j \leq N$, the ratio $\alpha_n^{(j)}/\alpha_n^{(1)}$ decays to zero by Assumption **(A:3)**(iii). Thus all coordinates except $x^{(1)}$ evolve on an *infinitely slower* clock and act, from the perspective of the $x^{(1)}$ dynamics, as *frozen* external parameters. Invoking the standard ODE method (Benaïm–Borkar framework) with the "third extension" of Chap. 2 in Borkar (2008) shows that the joint process tracks the flow of the differential inclusion

$$\dot{x}^{(1)}(t) = h^{(1)}\big(x^{(1)}(t), x^{(2)}(t), \ldots, x^{(N)}(t)\big), \quad \dot{x}^{(j)}(t) = 0 \ (2 \leq j \leq N),$$

whose internally chain–transitive (ICT) invariant set is $\big\{(\lambda^{(1)}(x^{(2)}, \ldots, x^{(N)}), x^{(2)}, \ldots, x^{(N)})\big\}$. Consequently $x_n^{(1)} - \lambda^{(1)}(x_n^{(2)}, \ldots, x_n^{(N)}) \to 0$ almost surely as $n \to \infty$.

3. **Cascade down the timescales.** We repeat the identical argument on the *next* stepsize sequence $\{\alpha_n^{(2)}\}$ while treating the remaining coordinates $(x^{(3)}, \ldots, x^{(N)})$ as quasi-static. Induction shows that for each $1 \leq k \leq N-1$

$$x_n^{(k)} - \lambda^{(k:N-1)}\big(x_n^{(N)}\big) \longrightarrow 0 \quad \text{a.s.}$$

where the iterated map $\lambda^{(k:N-1)}$ is the nested fixed-point operator from the theorem statement.

4. **ODE analysis on the slowest scale.** Introduce a continuous-time interpolation of the slowest iterate $x_n^{(N)}$ and analyse the martingale-error decomposition exactly as in Chap. 2, Lemma 2 of Borkar (2008). Gronwall's inequality bounds the difference between the interpolated trajectory and the solution of the limit ODE

$$\dot{x}^{(N)}(t) = h^{(N)}\Big(\lambda^{(1:N-1)}\big(x^{(N)}(t)\big), \ \ldots, \ \lambda^{(N-1)}\big(x^{(N)}(t)\big), \ x^{(N)}(t)\Big),$$

whose globally asymptotically stable equilibrium is $x_*^{(N)}$ by Assumption **(C.N.N)**. Hence $x_n^{(N)} \to x_*^{(N)}$ and, by Step 2, every faster coordinate converges to $\lambda^{(k:N-1)}(x_*^{(N)})$, completing the convergence proof *conditional on stability*. This yields Theorem 2.

5. **Bounding the iterates: a recursive scaling argument.** To prove almost-sure boundedness we compare the raw iterates with appropriately scaled versions that satisfy rescaled SA recursions. For the fastest scale one rescales by $r(n) \doteq \max\big(1, \|\bar{X}^{(1)}(T_n)\|\big)$; Lipschitz continuity of $h^{(1)}$ and the drift condition **(B.N.1)** force the scaled trajectory to contract whenever it exits a suitable multiple of the slower-scale coordinates. Formally, Lemma 3 shows that $\|x_n^{(1)}\| \leq K_1^* \sum_{j=2}^N \|x_n^{(j)}\|$.

   The identical construction applied to the $l$th timescale—now with the first $l-1$ coordinates treated as instantaneous and the rest as quasi-static—yields, by induction over $l = 2, \ldots, N-1$,

$$\|x_n^{(l)}\| \leq K_l^* \sum_{j=l+1}^N \|x_n^{(j)}\|.$$

   Finally, for the slowest iterate, one shows by contradiction that its norm cannot diverge; otherwise the scaled recursion would violate the one-step contraction inequality in Lemma 9. As a result all coordinates remain a.s. bounded, validating assumption **(B.N.N+1)** and establishing Theorem 3.

6. **Combining stability and convergence.** The boundedness of every coordinate proved in Step 5 feeds into Step 1–Step 3, thereby proving Theorem 1 in full generality.

**Remark 1.** *Since the additional error terms are $o(1)$, their contribution is asymptotically negligible. See arguments in the third extension of (Chapter 2, pp. 17 of Borkar (2008)) that handles this case for one-timescale iterates. Using similar arguments along with our analysis, this extension can be easily obtained.*

## 5 Application to Reinforcement Learning Algorithms

In this section we use our results on $N$-timescale recursions to show stability and convergence of two Reinforcement Learning (RL) algorithms: Gradient Temporal Difference (GTD) with momentum for policy evaluation, and Constrained Actor Critic for policy optimization in constrained RL.

In the standard RL framework, an agent interacts with a stochastic and dynamic environment. At each discrete time step $t$, the agent is in state $s_t \in \mathcal{S}$, picks an action $a_t \in \mathcal{A}$, receives a reward $r_{t+1} \equiv r(s_t, a_t, s_{t+1}) \in \mathcal{R}$ and probabilistically moves to another state $s_{t+1} \in \mathcal{S}$. The tuple $(\mathcal{S}, \mathcal{A}, \mathbb{P}, \mathcal{R}, \gamma)$ constitutes a Markov Decision Process (MDP). Here $\mathcal{S}$ and $\mathcal{A}$ are assumed finite. Also, $\gamma \in (0, 1)$ is the discount factor. A policy $\pi : \mathcal{S} \times \mathcal{A} \to [0, 1]$ is a mapping that defines the probability of picking an action in a state. We let $P^\pi(s'|s)$ denote the probability of transition to state $s'$ from state $s$ when an action is chosen as per $\pi$. We let $\{d^\pi(s)\}_{s \in \mathcal{S}}$ denote the steady-state distribution for the Markov chain induced by $\pi$. The matrix $D$ is a $n \times n$ diagonal matrix with elements $d^\pi(s)$ on its diagonals with $n$ being the number of states. The value function corresponding to state $s$ under policy $\pi$ for state $s$ is defined by:

$$V^\pi(s) = \mathbb{E}_\pi \left[ \sum_{t=0}^\infty \gamma^t R_{t+1} | s_0 = s \right].$$

### 5.1 Gradient Temporal Difference with Momentum

With linear function approximation for policy evaluation (i.e., for a fixed $\pi$), the goal is to estimate $V^\pi(s)$ from samples of the form $(s_t, r_{t+1}, s_{t+1})$ through a linear model $V_\theta(s) = \theta^T \phi(s)$. Here $\phi(s) \equiv \phi_s$ is a feature vector associated with the state $s$ and $\theta$ is the associated parameter vector. The TD-error is defined by $\delta_t = r_{t+1} + \gamma \theta_t^T \phi_{t+1} - \theta_t^T \phi_t$. The feature matrix $\Phi$ is an $n \times d$ matrix where the $s^{th}$ row is $\phi(s)^T$. In the following, we consider the i.i.d setting, where the tuple $(\phi_t, \phi_t')$ (with $\phi_t' \equiv \phi_{t+1}$) is drawn independently from the stationary distribution $\{d_\pi(s)\}$. Let $\bar{A} \triangleq \mathbb{E}[\phi_t(\gamma \phi_t' - \phi_t)^T]$ and $\bar{b} \triangleq \mathbb{E}[r_{t+1} \phi_t]$, where the expectations are w.r.t. the stationary distribution of the induced Markov chain. The matrix $\bar{A}$ is known to be negative definite (see Maei (2011); Tsitsiklis and Van Roy (1997)). In the off-policy setting, the behaviour policy $\pi$ is used to sample trajectories from the MDP while the target policy $\mu$ is the one whose associated value function needs to be approximated. Let $\rho_t = \frac{\pi(a_t|s_t)}{\mu(a_t|s_t)}$ denote the importance sampling ratio. Gradient TD algorithms are a class of TD algorithms that are convergent even in the off-policy setting. We first present the iterates associated with the algorithms GTD2 and TDC, see Sutton et al. (2009a).

1. **GTD2**:

$$\theta_{t+1} = \theta_t + \alpha_t(\phi_t - \gamma \phi_t')\phi_t^T u_t, \tag{7}$$

$$u_{t+1} = u_t + \beta_t(\delta_t - \phi_t^T u_t)\phi_t. \tag{8}$$

2. **TDC**:

$$\theta_{t+1} = \theta_t + \alpha_t \delta_t \phi_t - \alpha_t \gamma \phi_t'(\phi_t^T u_t), \tag{9}$$

$$u_{t+1} = u_t + \beta_t(\delta_t - \phi_t^T u_t)\phi_t. \tag{10}$$

In the GTD algorithm, the objective function considered is the Norm of Expected Error defined as $NEU(\theta) = \mathbb{E}[\delta\phi]$ and the algorithm is derived by expressing the gradient direction as $-\frac{1}{2}\nabla NEU(\theta) = \mathbb{E}\left[(\phi - \gamma\phi')\phi^T\right]\mathbb{E}[\delta(\theta)\phi]$. Here $\phi' \equiv \phi(s')$. Since the expectation becomes biased by the correlation of the two terms if both the terms are sampled separately, an estimate of the second expectation is maintained as a long-term quasi-stationary estimate while samples for the first expectation are used. For GTD2 and TDC, a similar approach is used on the objective function Mean Square Projected Bellman Error defined as $MSPBE(\theta) = \|V_\theta - \Pi T^\pi V_\theta\|_D$, where for any $x \in \mathbb{R}^n$, $\|x\|_D = \sqrt{x^T D x}$. Here, $\Pi$ is the projection operator that projects vectors to the subspace $\{\Phi\theta|\theta \in \mathbb{R}^d\}$ and $T^\pi$ is the Bellman operator defined as $T^\pi V = R^\pi + \gamma P^\pi V$. It was shown in all the three cases that $\theta_n \to \theta^* = -\bar{A}^{-1}\bar{b}$.

### 5.1.1 Three Timescale Gradient TD Algorithms with Momentum

We consider the Gradient TD algorithms with an added heavy ball term to the first iterate.

1. GTD2 with momentum (**GTD2-M-3TS**):

$$\theta_{t+1} = \theta_t + \alpha_t(\phi_t - \gamma\phi'_t)\phi_t^T u_t + \eta_t(\theta_t - \theta_{t-1}), \tag{11}$$

$$u_{t+1} = u_t + \beta_t(\delta_t - \phi_t^T u_t)\phi_t. \tag{12}$$

2. TDC with momentum (**TDC-M-3TS**):

$$\theta_{t+1} = \theta_t + \alpha_t(\delta_t\phi_t - \gamma\phi'_t(\phi_t^T u_t)) + \eta_t(\theta_t - \theta_{t-1}), \tag{13}$$

$$u_{t+1} = u_t + \beta_t(\delta_t - \phi_t^T u_t)\phi_t. \tag{14}$$

The momentum parameter $\eta_t$ is chosen as in Avrachenkov et al. (2020) as $\eta_t = \frac{\varrho_t - w\alpha_t}{\varrho_{t-1}}$, where $\{\varrho_t\}$ is a positive sequence and $w \in \mathbb{R}$ is a constant. We let $\frac{\theta_{t+1}-\theta_t}{\varrho_t} = v_{t+1}, \xi_t = \frac{\alpha_t}{\varrho_t}$ and $\varepsilon_t = v_{t+1} - v_t$. Then the iterates for **GTD2-M-3TS** can then be decomposed into the three recursions as below:

$$v_{t+1} = v_t + \xi_t\left((\phi_t - \gamma\phi'_t)\phi_t^T u_t - wv_t\right), \tag{15}$$

$$u_{t+1} = u_t + \beta_t(\delta_t\phi_t - \phi_t\phi_t^T u_t), \tag{16}$$

$$\theta_{t+1} = \theta_t + \varrho_t(v_t + \varepsilon_t). \tag{17}$$

Similarly, the iterates for **TDC-M-3TS** can be decomposed as:

$$v_{t+1} = v_t + \xi_t\left(\delta_t\phi_t - \gamma\phi'_t\phi_t^T u_t - wv_t\right), \tag{18}$$

$$u_{t+1} = u_t + \beta_t(\delta_t\phi_t - \phi_t\phi_t^T u_t), \tag{19}$$

$$\theta_{t+1} = \theta_t + \varrho_t(v_t + \varepsilon_t). \tag{20}$$

Consider the following assumptions:

**Assumption 1.** *All rewards $r(s, s')$ and features $\phi(s)$ are bounded, i.e., $r(s, s') \leq 1$ and $\|\phi(s)\| \leq 1 \ \forall s, s' \in \mathcal{S}$. Also, the matrix $\Phi$ has full rank, where $\Phi$ is an $n \times d$ matrix where the $s^{th}$ row is $\phi(s)^T$.*

**Assumption 2.** *The step-sizes satisfy $\xi_t > 0, \beta_t > 0, \varrho_t > 0 \ \forall t,$*

$$\sum_t \xi_t = \sum_t \beta_t = \sum_t \varrho_t = \infty, \ \ \sum_t(\xi_t^2 + \beta_t^2 + \varrho_t^2) < \infty,$$

$\frac{\beta_t}{\xi_t} \to 0, \frac{\varrho_t}{\beta_t} \to 0$ *as $t \to \infty$, and the momentum parameter satisfies: $\eta_t = \frac{\varrho_t - w\alpha_t}{\varrho_{t-1}}.$*

**Assumption 3.** *The samples $(\phi_t, \phi'_t)$ are drawn i.i.d from the stationary distribution of the Markov chain induced by the target policy $\pi$.*

**Remark 2.** *Assumptions 1-2 are standard requirements. Assumption 3, on the other hand, is a restrictive requirement though often used in the literature, see for instance, Maei (2011); Sutton et al. (2009a;b); Dalal et al. (2018a; 2020), where this assumption has been made. This can however be relaxed if our analysis of N-timescale SA algorithms is extended to the case when the noise can have a general iterate-dependent Markovian structure that can appear in each of the N recursions. This setting is popularly referred to as the Markov noise setting. There is no prior work on stability of multi-scale algorithms with Markov noise even though Ramaswamy and Bhatnagar (2019) does present stability conditions for one-timescale algorithms with Markov noise. We say more on this in Section 6.*

**Theorem 5.** *Suppose Assumptions 1, 2 and 3 hold and let $w > 0$. Then, the GTD2-M-3TS iterates given by equation 15-equation 17 satisfy $\theta_t \to \theta^* = -\bar{A}^{-1}\bar{b}$ a.s. as $t \to \infty$.*

*Proof.* We transform equation 15-equation 17 into the standard form. Let $\mathcal{F}_t = \sigma(u_0, v_0, \theta_0, r_{j+1}, \phi_j, \phi'_j : j < t)$. Let, $A_t = \phi_t(\gamma\phi'_t - \phi_t)^T$ and $b_t = r_{t+1}\phi_t$. Then, equation 15 can be rewritten as:

$$v_{t+1} = v_t + \xi_t\left(h^{(1)}(v_t, u_t, \theta_t) + M^{(1)}_{t+1}\right)$$

$$\text{where, } h^{(1)}(v_t, u_t, \theta_t) = \mathbb{E}[(\phi_t - \gamma\phi'_t)\phi_t^T u_t - wv_t|\mathcal{F}_t] = -\bar{A}^T u_t - wv_t,$$

$$M^{(1)}_{t+1} = -A_t^T u_t - wv_t - h(v_t, u_t, \theta_t) = (\bar{A}^T - A_t^T)u_t.$$

Next, equation 16 can be re-written as:

$$u_{t+1} = u_t + \beta_t\left(h^{(2)}(v_t, u_t, \theta_t) + M^{(2)}_{t+1}\right),$$

$$\text{where, } h^{(2)}(v_t, u_t, \theta_t) = \mathbb{E}[\delta_t\phi_t - \phi_t\phi_t^T u_t|\mathcal{F}_t] = \bar{A}\theta_t + \bar{b} - \bar{C}u_t,$$

$$M^{(2)}_{t+1} = A_t\theta_t + b_t - C_t u_t - g(v_t, u_t, \theta_t), = (A_t - \bar{A})\theta_t + (b_t - \bar{b}) + (\bar{C} - C_t)u_t.$$

Here, $C_t = \phi_t\phi_t^T$ and $\bar{C} = \mathbb{E}[\phi_t\phi_t^T]$. Finally, equation 17 can be re-written as:

$$\theta_{t+1} = \theta_t + \varrho_t\left(h^{(3)}(v_t, u_t, \theta_t) + \varepsilon_t + M^{(3)}_{t+1}\right),$$

where $h^{(3)}(v_t, u_t, \theta_t) = v_t$ and $M^{(3)}_{t+1} = 0$. We show that conditions **(A:1)**-**(A:3)**, **(B.3.1)**-**(B.3.3)** and **(C.3.1)**-**(C.3.3)** hold. The functions $h^{(1)}, h^{(2)}, h^{(3)}$ are linear in $v, u, \theta$ and hence Lipschitz continuous, thereby satisfying **(A:1)**. We choose the step-size sequences such that they satisfy **(A:2)**. One popular choice is

$$\xi_t = \frac{1}{(t+1)^\xi}, \beta_t = \frac{1}{(t+1)^\beta}, \varrho_t = \frac{1}{(t+1)^\varrho}, \tag{21}$$

with $\frac{1}{2} < \xi < \beta < \varrho \le 1$. Now, $M^{(1)}_{t+1}, M^{(2)}_{t+1}$ and $M^{(3)}_{t+1}$ $t \ge 0$, are martingale difference sequences w.r.t $\mathcal{F}_t$ by construction. Further,

$$\mathbb{E}[\|M^{(1)}_{t+1}\|^2|\mathcal{F}_t] \le \|(\bar{A}^T - A_t^T)\|^2\|u_t\|^2,$$

$$\mathbb{E}[\|M^{(2)}_{t+1}\|^2|\mathcal{F}_t] \le 3(\|(A_t - \bar{A})\|^2\|\theta_t\|^2 + \|(b_t - \bar{b})\|^2 + \|(\bar{C} - C_t)\|^2\|u_t\|^2).$$

Note that **(A:3)** is satisfied with $K_1 = \|(\bar{A}^T - A_t^T)\|^2$, $K_2 = 3\max(\|A_t - \bar{A}\|^2, \|b_t - \bar{b}\|^2, \|(\bar{C} - C_t)\|^2)$, and any $K_3 > 0$. The fact that $K_1, K_2 < \infty$ follows from Assumption 1. For a fixed $u, \theta \in \mathbb{R}^d$, consider the ODE

$$\dot{v}(t) = -\bar{A}^T u - wv(t).$$

For $w > 0$, $\lambda^{(1)}(u, \theta) = -\frac{\bar{A}^T u}{w}$ is the unique globally asymptotically stable equilibrium (g.a.s.e), is linear and therefore Lipschitz continuous. This satisfies **(C.3.1)**. Next, for a fixed $\theta \in \mathbb{R}^d$,

$$\dot{u}(t) = \bar{A}\theta + \bar{b} - \bar{C}u(t),$$

has $\lambda^{(2)}(\theta) = \bar{C}^{-1}(\bar{A}\theta + \bar{b})$ as its unique g.a.s.e because $-\bar{C}^{-1}$ is negative definite. Also $\lambda^{(2)}(\theta)$ is linear in $\theta$ and therefore Lipschitz. This satisfies **(C.3.2)**. Finally, to satisfy **(C.3.3)**, consider,

$$\dot{\theta}(t) = \frac{-\bar{A}^T\bar{C}^{-1}\bar{A}\theta(t) - \bar{A}^T\bar{C}^{-1}\bar{b}}{w}.$$

Since $\bar{A}$ is negative definite and $\bar{C}$ is positive definite, $-\bar{A}^T\bar{C}^{-1}\bar{A}$ is negative definite as well. Therefore, $\theta^* = -\bar{A}^{-1}\bar{b}$ is the unique g.a.s.e for the above ODE.

Next, we show that the sufficient conditions for stability of the three iterates are satisfied. The function, $h_c^{(1)}(v, u, \theta) = \frac{-c\bar{A}^T u - wcv}{c} = -\bar{A}^T u - wv \to h_\infty^{(1)}(v, u, \theta) = -\bar{A}^T u - wv$ uniformly on compacts as $c \to \infty$. The limiting ODE:

$$\dot{v}(t) = -\bar{A}^T u - wv(t)$$

has $\lambda_\infty^{(1)}(u, \theta) = -\frac{\bar{A}^T u}{w}$ as its unique g.a.s.e. $\lambda_\infty^{(1)}$ is Lipschitz with $\lambda_\infty^{(1)}(0, 0) = 0$, thus satisfying assumption **(B.3.1)** The function, $h_c^{(2)}(u, \theta) = \frac{c\bar{A}\theta + \bar{b} - c\bar{C}u}{c} = \bar{A}\theta - \bar{C}u + \frac{\bar{b}}{c} \to h_\infty^{(2)}(u, \theta) = \bar{A}\theta - \bar{C}u$ uniformly on compacts as $c \to \infty$. The limiting ODE

$$\dot{u}(t) = \bar{A}\theta - \bar{C}u(t)$$

has $\lambda_\infty^{(2)}(\theta) = \bar{C}^{-1}\bar{A}\theta$ as its unique g.a.s.e. since $-\bar{C}$ is negative definite. $\lambda_\infty^{(2)}$ is Lipschitz with $\lambda_\infty^{(2)}(0) = 0$. Thus assumption **(B.3.2)** is satisfied.

Finally, $h_c^{(3)}(\theta) = \frac{-c\bar{A}^T\bar{C}^{-1}\bar{A}\theta}{cw} \to h_\infty^{(3)}(\theta) = \frac{-\bar{A}^T\bar{C}^{-1}\bar{A}\theta}{w}$ uniformly on compacts as $c \to \infty$ and the ODE:

$$\dot{\theta}(t) = -\frac{\bar{A}^T\bar{C}^{-1}\bar{A}\theta(t)}{w}$$

has the origin in $\mathbb{R}^d$ as its unique g.a.s.e. This ensures the final condition **(B.3.3)**. Further, observe that $\|\varepsilon_t^{(3)}\| = \xi_t \| \left( (\phi_t - \gamma\phi_t')\phi_t^T u_t - wv_t \right) \| \to 0$ since $\xi_t \to 0$ as $t \to \infty$. By Theorem 4,

$$\begin{pmatrix} v_t \\ u_t \\ \theta_t \end{pmatrix} \to \begin{pmatrix} \lambda(\Gamma(-\bar{A}^{-1}\bar{b}), -\bar{A}^{-1}\bar{b}) \\ \Gamma(-\bar{A}^{-1}\bar{b}) \\ -\bar{A}^{-1}\bar{b}. \end{pmatrix} = \begin{pmatrix} 0 \\ 0 \\ -\bar{A}^{-1}\bar{b}. \end{pmatrix}$$

Specifically, $\theta_t \to -\bar{A}^{-1}\bar{b}$. □

**Theorem 6.** *Suppose Assumptions 1, 2 and 3 hold and let $w > 0$. Then, the TDC-M-3TS iterates given by equation 18–equation 20 satisfy $\theta_t \to \theta^* = -\bar{A}^{-1}\bar{b}$ a.s. as $t \to \infty$.*

*Proof.* As before, we transform the iterates given by equation 18, equation 19 and equation 20 into the standard SA form. Let $\mathcal{F}_t = \sigma(u_0, v_0, \theta_0, r_{j+1}, \phi_j, \phi_j' : j < t)$. Let $A_t = \phi_t(\gamma\phi_t' - \phi_t)^T$ and $b_t = r_{t+1}\phi_t$. Then, equation 18 can be re-written as:

$$v_{t+1} = v_t + \xi_t \left( h^{(1)}(v_t, u_t, \theta_t) + M_{t+1}^{(1)} \right),$$

where $h^{(1)}(v_t, u_t, \theta_t) = \mathbb{E}[\delta_t\phi_t - \gamma\phi_t'\phi_t^T u_t - wv_t | \mathcal{F}_t] = \bar{A}\theta_t + \bar{b} - \gamma\mathbb{E}[\phi_t'\phi_t^T]u_t - wv_t$ and $M_{t+1}^{(1)} = \delta_t\phi_t - \gamma\phi_t'\phi_t^T u_t - wv_t - h^{(1)}(v_t, u_t, \theta_t) = (A_t - \bar{A})\theta_t + (b_t - \bar{b}) + \gamma(\mathbb{E}[\phi_t'\phi_t^T] - \phi_t'\phi_t^T)u_t$. Next, equation 19 can be re-written as:

$$u_{t+1} = u_t + \beta_t \left( h^{(2)}(v_t, u_t, \theta_t) + M_{t+1}^{(2)} \right)$$

where, $h^{(2)}(v_t, u_t, \theta_t) = \mathbb{E}[\delta_t\phi_t - \phi_t\phi_t^T u_t | \mathcal{F}_t] = \bar{A}\theta_t + \bar{b} - \bar{C}u_t$ and $M_{t+1}^{(2)} = A_t\theta_t + b_t - C_t u_t - h^{(2)}(v_t, u_t, \theta_t) = (A_t - \bar{A})\theta_t + (b_t - \bar{b}) + (\bar{C} - C_t)u_t$. Here, $C_t = \phi_t\phi_t^T$, $\bar{C} = \mathbb{E}[\phi_t\phi_t^T]$. Finally, equation 20 can be re-written as:

$$\theta_{t+1} = \theta_t + \varrho_t \left( h^{(3)}(v_t, u_t, \theta_t) + \varepsilon_t + M_{t+1}^{(3)} \right),$$

where $h^{(3)}(v_t, u_t, \theta_t) = v_t$ and $M_{t+1}^{(3)} = 0$. As before we show that the conditions **(A:1)-(A:3)**, **(B.3.1)-(B.3.3)** and **(C.3.1)-(C.3.3)** hold. The functions $h^{(1)}, h^{(2)}, h^{(3)}$ are linear in $v, u, \theta$ and hence Lipschitz continuous, therefore satisfying **(A:1)**. We choose the step-size sequences such that they satisfy **(A:2)**. One popular choice is equation 21. Observe now that $M_{t+1}^{(1)}, M_{t+1}^{(2)}$ and $M_{t+1}^{(3)}$ $t \geq 0$, are martingale difference sequences w.r.t $\mathcal{F}_t$ by construction. Next,

$$\mathbb{E}[\|M_{t+1}^{(1)}\|^2 | \mathcal{F}_t] \leq 3(\|(A_t - \bar{A})\|^2\|\theta_t\|^2 + \|(b_t - \bar{b})\|^2 + \gamma(\|\mathbb{E}[\phi_t'\phi_t^T] - \phi_t'\phi_t^T\|^2)\|u_t\|^2),$$

$$\mathbb{E}[\|M_{t+1}^{(2)}\|^2|\mathcal{F}_t] \leq 3(\|(A_t - \bar{A})\|^2\|\theta_t\|^2 + \|(b_t - \bar{b})\|^2 + \|(\bar{C} - C_t)\|^2\|u_t\|^2).$$

The first part of (**A:3**) is satisfied with $K_1 = 3\max(\|(A_t - \bar{A})\|^2, \|(b_t - \bar{b})\|^2, \gamma(\|\mathbb{E}[\phi_t'\phi_t^T] - \phi_t'\phi_t^T\|^2)), K_2 = 3\max(\|A_t - \bar{A}\|^2, \|b_t - \bar{b}\|^2, \|(\bar{C} - C_t)\|^2)$ and any $K_3 > 0$. The fact that $K_1, K_2 < \infty$ follows from Assumption 1. For a fixed $u, \theta \in \mathbb{R}^d$, consider the ODE

$$\dot{v}(t) = \bar{A}\theta + \bar{b} - \gamma\mathbb{E}[\phi_t'\phi_t^T]u - wv(t).$$

For $w > 0$, $\lambda^{(1)}(u, \theta) = \frac{\bar{A}\theta + \bar{b} - \gamma\mathbb{E}[\phi_t'\phi_t^T]u}{w}$ is the unique g.a.s.e, is linear and therefore Lipschitz continuous. This satisfies (**C.3.1**). Next, for a fixed $\theta \in \mathbb{R}^d$,

$$\dot{u}(t) = \bar{A}\theta + \bar{b} - \bar{C}u(t),$$

has $\lambda^{(2)}(\theta) = \bar{C}^{-1}(\bar{A}\theta + \bar{b})$ as its unique g.a.s.e because $-\bar{C}^{-1}$ is negative definite. Also $\lambda^{(2)}(\theta)$ is linear in $\theta$ and therefore Lipschitz. This satisfies (**C.3.2**). Finally, to satisfy (**C.3.3**), consider,

$$\dot{\theta}(t) = \frac{(I - \gamma\mathbb{E}[\phi_t'\phi_t^T]\bar{C}^{-1})(\bar{A}\theta(t) + \bar{b})}{w}.$$

Now, $(I - \gamma\mathbb{E}[\phi_t'\phi_t^T]\bar{C}^{-1})\bar{A} = (\mathbb{E}[\phi_t\phi_t^T] - \gamma\mathbb{E}[\phi_t'\phi_t^T])\bar{C}^{-1}\bar{A} = \mathbb{E}[(\phi_t - \gamma\phi_t')\phi_t^T]\bar{C}^{-1}\bar{A} = -\bar{A}^T\bar{C}^{-1}\bar{A}$. Since, $\bar{A}$ is negative definite and $\bar{C}$ is positive definite, therefore $-\bar{A}^T\bar{C}^{-1}\bar{A}$ is negative definite and hence the above ODE has $\theta^* = -\bar{A}^{-1}\bar{b}$ as its unique g.a.s.e.

Next, we show that the sufficient conditions for stability of the three iterates are satisfied. The function, $h_c^{(1)}(v, u, \theta) = \frac{c\bar{A}\theta + \bar{b} - c\gamma\mathbb{E}[\phi_t'\phi_t^T]u - cwv}{c} = \bar{A}\theta + \bar{b}c - \gamma\mathbb{E}[\phi_t'\phi_t^T]u - wv \to h_\infty^{(1)}(v, u, \theta) = \bar{A}\theta - \gamma\mathbb{E}[\phi_t'\phi_t^T]u - wv$ uniformly on compacts as $c \to \infty$. The limiting ODE:

$$\dot{v}(t) = \bar{A}\theta_t - \gamma\mathbb{E}[\phi_t'\phi_t^T]u_t - wv(t)$$

has $\lambda_\infty^{(1)}(u, \theta) = \frac{\bar{A}\theta - \gamma\mathbb{E}[\phi_t'\phi_t^T]u}{w}$ as its unique g.a.s.e. $\lambda_\infty^{(1)}$ is Lipschitz with $\lambda_\infty^{(1)}(0, 0) = 0$, thus satisfying assumption (**B.3.1**).

The function $h_c^{(2)}(u, \theta) = \frac{c\bar{A}\theta + \bar{b} - c\bar{C}u}{c} = \bar{A}\theta - \bar{C}u + \frac{\bar{b}}{c} \to h_\infty^{(2)}(u, \theta) = \bar{A}\theta - \bar{C}u$ uniformly on compacts as $c \to \infty$. The limiting ODE

$$\dot{u}(t) = \bar{A}\theta - \bar{C}u(t)$$

has $\lambda_\infty^{(2)}(\theta) = \bar{C}^{-1}\bar{A}\theta$ as its unique g.a.s.e. since $-\bar{C}$ is negative definite. $\lambda_\infty^{(2)}$ is Lipschitz with $\lambda_\infty^{(2)}(0) = 0$. Thus assumption (**B.3.2**) is satisfied.

Finally,

$$h_c^{(3)}(\theta) = \frac{c\bar{A}\theta - c\gamma\mathbb{E}[\phi_t'\phi_t^T]\bar{C}^{-1}\bar{A}\theta}{cw} \to h_\infty^{(3)} = \frac{(I - \gamma\mathbb{E}[\phi_t'\phi_t^T]\bar{C}^{-1})\bar{A}\theta}{w}$$

uniformly on compacts as $c \to \infty$. Consider the ODE:

$$\dot{\theta}(t) = \frac{(I - \gamma\mathbb{E}[\phi_t'\phi_t^T]\bar{C}^{-1})\bar{A}\theta(t)}{w}.$$

Now $(I - \gamma\mathbb{E}[\phi_t'\phi_t^T]\bar{C}^{-1})\bar{A} = (\mathbb{E}[\phi_t\phi_t^T] - \gamma\mathbb{E}[\phi_t'\phi_t^T])\bar{C}^{-1}\bar{A} = \mathbb{E}[(\phi_t - \gamma\phi_t')\phi_t^T]\bar{C}^{-1}\bar{A} = -\bar{A}^T\bar{C}^{-1}\bar{A}$. Since $\bar{A}$ is negative definite and $\bar{C}$ is positive definite, $-\bar{A}^T\bar{C}^{-1}\bar{A}$ is negative definite and hence the above ODE has the origin as its unique g.a.s.e. This ensures the final condition (**B.3.3**). Next, observe that $\|\varepsilon_t^{(3)}\| = \xi_t\|\left((\phi_t - \gamma\phi_t')\phi_t^T u_t - wv_t\right)\| \to 0$ since $\xi_t \to 0$ as $t \to \infty$. By Theorem 4,

$$\begin{pmatrix} v_t \\ u_t \\ \theta_t \end{pmatrix} \to \begin{pmatrix} \lambda(\Gamma(-\bar{A}^{-1}\bar{b}), -\bar{A}^{-1}\bar{b}) \\ \Gamma(-\bar{A}^{-1}\bar{b}) \\ -\bar{A}^{-1}\bar{b} \end{pmatrix} = \begin{pmatrix} 0 \\ 0 \\ -\bar{A}^{-1}\bar{b} \end{pmatrix}.$$

Specifically, $\theta_t \to -\bar{A}^{-1}\bar{b}$ almost surely.                    $\square$

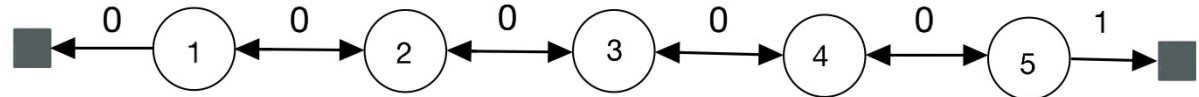

Figure 1: 5-State Random Walk from Sutton et al. (2009a).

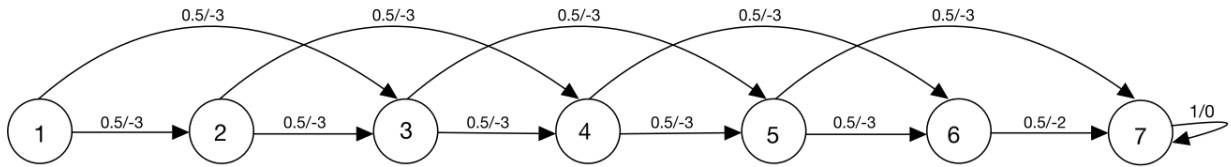

Figure 2: 7-state Boyan Chain from Boyan (1999)

#### 5.1.2 Experiments

The Gradient TD algorithms along with their momentum variants are evaluated on two standard MDPs: 5-State Random Walk Sutton et al. (2009a) and Boyan Chain Boyan (1999). Along with 3-timescale version of the algorithms we consider a 4-timescale version as defined by the following iterates:

1. **GTD2-M-4TS**:

$$\theta_{t+1} = \theta_t + \alpha_t(\phi_t - \gamma\phi'_t)\phi_t^T u_t + \eta_t^{(1)}(\theta_t - \theta_{t-1}), \tag{22}$$

$$u_{t+1} = u_t + \beta_t(\delta_t - \phi_t^T u_t)\phi_t + \eta_t^{(2)}(u_t - u_{t-1}). \tag{23}$$

2. **TDC-M-4TS**:

$$\theta_{t+1} = \theta_t + \alpha_t(\delta_t\phi_t - \gamma\phi'_t(\phi_t^T u_t)) + \eta_t^{(1)}(\theta_t - \theta_{t-1}), \tag{24}$$

$$u_{t+1} = u_t + \beta_t(\delta_t - \phi_t^T u_t)\phi_t + \eta_t^{(2)}(u_t - u_{t-1}). \tag{25}$$

We consider decreasing step-sizes of the form: $\varrho_t^{(1)} = \frac{1}{(t+1)^{\varrho^{(1)}}}$, $\varrho_t^{(2)} = \frac{1}{(t+1)^{\varrho^{(2)}}}$, $\beta_t = \frac{1}{(t+1)^{\beta}}$, $\alpha_t = \frac{1}{(t+1)^{\alpha}}$, respectively, in all the examples. In the 3-Timescale case the conditions on step size turn out to be $\alpha < \varrho + \beta$, and $\beta < \varrho$, while in the 4-Timescale case the conditions are $\alpha < \beta + \varrho^{(1)} - \varrho^{(2)}$, $\beta < 2\varrho^{(2)}$ and $\varrho^{(2)} < \varrho^{(1)}$. Our analysis of convergence requires square-summability of step-size sequences. However, such a choice is seen to slow down the convergence of the algorithm. Recently, Dalal et al. (2018a) provided convergence rate results for Gradient TD schemes with non-square-summable step-sizes (See Remark 2 of Dalal et al. (2018a)). Motivated by this, we look at non-square summable step-sizes for our experiments, and observe that the iterates empirically converge in such cases as well.

For a detailed description of the MDPs, see Figure 1 and 2. See Figure 3 for the results on the 5-State Random Walk and Figure 4 for results on the Boyan Chain. The exact values of $\varrho^{(1)}$, $\varrho^{(2)}$, $\alpha$ and $\beta$ are provided in Table 1 and $w = 0.1$ for both 3-TS and 4-TS settings. Our results indicate that adding momentum terms to the algorithms clearly improves performance over their vanilla counterparts. Nonetheless, it is less clear from these experiments as to whether adding momentum term to both the iterates is better as opposed to doing the same for only one of the iterates. In particular, for GTD2, the 4-TS scheme appears to perform better while for TDC, the 3-TS version is seen to be better. Further analysis of the finite time behaviour of these algorithms needs to be carried out in the future to better assess the performance of these algorithms.

Table 1: Choice of step-size parameters

| 5-State RW | $\alpha$ | $\beta$ | $\varrho^{(1)}$ | $\varrho^{(2)}$ |
|---|---|---|---|---|
| Vanilla | 0.4 | 0.4 | - | - |
| Three-TS | 0.4 | 0.4 | 0.5 | - |
| Four-TS | 0.4 | 0.4 | 0.5 | 0.25 |
| Boyan Chain | $\alpha$ | $\beta$ | $\varrho^{(1)}$ | $\varrho^{(2)}$ |
| Vanilla | 0.4 | 0.4 | - | - |
| Three-TS | 0.35 | 0.35 | 0.45 | - |
| Four-TS | 0.35 | 0.35 | 0.45 | 0.35 |

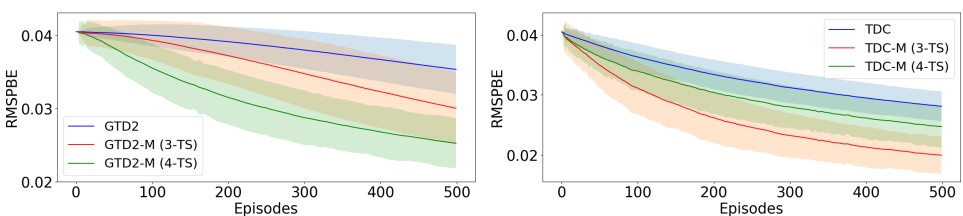

Figure 3: RMSPBE across episodes (averaged over 100 independent runs) for 5-State Random Walk. The features used are the *Dependent* features used in Sutton et al. (2009a).

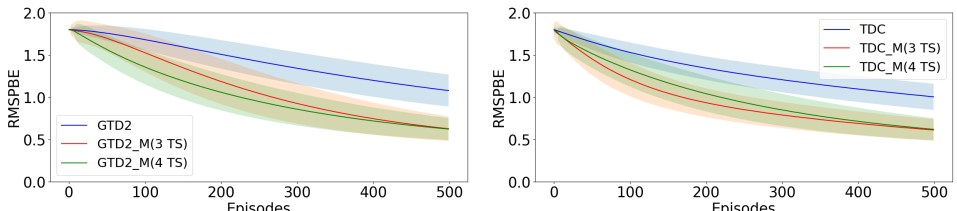

Figure 4: RMSPBE (averaged over 100 independent runs) across episodes for the Boyan Chain problem. The features used are the standard spiked features of size 4 used in Boyan chain (see Dann et al. (2014)).

## 5.2 Off-Policy Actor-Critic

We consider policy optimization with linear function approximation for policy evaluation. Let $\phi(s) \equiv \phi_s \in \mathbb{R}^{d_1}$ denote the critic feature associated with state $s$, and let $\theta \in \mathbb{R}^{d_1}$ denote the corresponding critic parameter. We work in the discounted cost-minimization setting, where the objective is to find a policy $\pi^*$ minimizing

$$J^\pi = \mathbb{E}_{s_0 \sim \xi} \Big[ \sum_{i=0}^{\infty} \gamma^i c_i \,\Big|\, s_0 \Big],$$

with $\xi$ denoting the initial-state distribution. We consider the following off-policy actor-critic recursion:

$$u_{t+1} = u_t + \alpha_t \rho_t \big( \delta_t - \phi_t^\top u_t \big) \phi_t, \tag{26}$$

$$\theta_{t+1} = \theta_t + \beta_t \rho_t \Big( \delta_t \phi_t - \gamma \phi_t' (\phi_t^\top u_t) \Big), \tag{27}$$

$$z_{t+1} = z_t - \eta_t \Big( \rho_t \delta_t \nabla_{z_t} \log \pi_{z_t}(a_t|s_t) + \epsilon z_t \Big), \tag{28}$$

where $\phi_t \equiv \phi(s_t)$, $\phi_t' \equiv \phi(s_{t+1})$,

$$\delta_t = c_{t+1} + \gamma \theta_t^\top \phi_t' - \theta_t^\top \phi_t, \qquad \rho_t = \frac{\pi_{z_t}(a_t|s_t)}{b(a_t|s_t)},$$

and $b(a|s)$ is the behavior policy used to generate samples. The additional term $\epsilon z_t$, with $\epsilon > 0$, corresponds to adding the regularization term $\frac{\epsilon}{2}\|z\|^2$ to the actor objective. These updates can also be interpreted as a regularized three-timescale analogue of those studied in Degris et al. (2012).

We assume throughout that the actor is parameterized by the Gibbs policy

$$\pi_z(a|s) = \frac{e^{\zeta(s,a)^\top z}}{\sum_{a'} e^{\zeta(s,a')^\top z}},$$

where $\zeta(s,a) \in \mathbb{R}^{d_2}$ denotes the actor feature vector. In particular,

$$\nabla_z \log \pi_z(a|s) = \zeta(s,a) - \sum_{a'} \zeta(s,a')\pi_z(a'|s).$$

For the analysis, we work in the i.i.d. sampling model, in which the tuples $(s_t, a_t, c_{t+1}, s_{t+1})$ are drawn independently according to the behavior policy and its stationary state distribution $d_b$. Let $\Phi$ denote the critic feature matrix whose $s$-th row is $\phi(s)^\top$. We impose the following assumptions.

**Assumption 4.** *The one-step costs and the critic and actor features are uniformly bounded, i.e.,*

$$|c(s,s')| \leq 1, \qquad \|\phi(s)\| \leq 1, \qquad \|\zeta(s,a)\| \leq 1$$

*for all $s, s' \in \mathcal{S}$ and $a \in \mathcal{A}$. Moreover, $\Phi$ has full column rank.*

**Assumption 5.** *The stationary distribution $d_b$ of the behavior policy has full support, i.e.,*

$$d_b(s) > 0, \qquad \forall s \in \mathcal{S}.$$

**Assumption 6.** *The behavior policy is uniformly positive: there exists $b_{\min} \in (0,1]$ such that*

$$b(a|s) \geq b_{\min}, \qquad \forall (s,a) \in \mathcal{S} \times \mathcal{A}.$$

**Assumption 7.** *The step-size sequences satisfy*

$$\alpha_t > 0, \quad \beta_t > 0, \quad \eta_t > 0, \qquad \sum_t \alpha_t = \sum_t \beta_t = \sum_t \eta_t = \infty,$$

$$\sum_t (\alpha_t^2 + \beta_t^2 + \eta_t^2) < \infty, \qquad \frac{\beta_t}{\alpha_t} \to 0, \qquad \frac{\eta_t}{\beta_t} \to 0.$$

*A standard choice is*

$$\alpha_t = \frac{1}{(t+1)^a}, \qquad \beta_t = \frac{1}{(t+1)^b}, \qquad \eta_t = \frac{1}{(t+1)^c},$$

*with $\frac{1}{2} < c < b < a \leq 1$.*

We now rewrite the recursion in the form required by the general three-timescale framework. Let

$$\mathcal{F}_t = \sigma(z_0, u_0, \theta_0, c_{j+1}, \phi_j, \phi_j' : j < t),$$

and define

$$A_t = \rho_t \phi_t (\gamma \phi_t' - \phi_t)^\top, \qquad b_t = \rho_t c_{t+1} \phi_t, \qquad C_t = \rho_t \phi_t \phi_t^\top.$$

Further, let

$$\bar{A}_z \triangleq \mathbb{E}[A_t], \qquad \bar{b}_z \triangleq \mathbb{E}[b_t], \qquad \bar{C} \triangleq \mathbb{E}[C_t].$$

Since $\phi_t$ depends only on $s_t$, we have

$$\bar{C} = \sum_s d_b(s) \sum_a b(a|s)\frac{\pi_z(a|s)}{b(a|s)}\phi(s)\phi(s)^\top = \sum_s d_b(s)\phi(s)\phi(s)^\top = \Phi^\top D_b \Phi,$$

where $D_b = \text{diag}(d_b)$. Hence $\bar{C}$ does not depend on $z$. By Assumptions 4 and 5, $\bar{C}$ is positive definite. The recursion can therefore be written as

$$u_{t+1} = u_t + \alpha_t\big(h^{(1)}(u_t, \theta_t, z_t) + M_{t+1}^{(1)}\big), \tag{29}$$

$$\theta_{t+1} = \theta_t + \beta_t\big(h^{(2)}(u_t, \theta_t, z_t) + M_{t+1}^{(2)}\big), \tag{30}$$

$$z_{t+1} = z_t + \eta_t\big(h^{(3)}(u_t, \theta_t, z_t) + M_{t+1}^{(3)}\big), \tag{31}$$

where

$$h^{(1)}(u, \theta, z) = \mathbb{E}\left[\rho_t(\delta_t - \phi_t^\top u)\phi_t | \mathcal{F}_t\right] = \bar{A}_z\theta + \bar{b}_z - \bar{C}u,$$

$$h^{(2)}(u, \theta, z) = \mathbb{E}\left[\rho_t(\delta_t\phi_t - \gamma\phi_t'\phi_t^\top u) | \mathcal{F}_t\right] = \bar{A}_z\theta + \bar{b}_z - \gamma\bar{G}_z u,$$

$$h^{(3)}(u, \theta, z) = -\mathbb{E}\left[\rho_t\delta_t\nabla_z \log \pi_z(a_t|s_t) | \mathcal{F}_t\right] - \epsilon z,$$

with

$$\bar{G}_z \triangleq \mathbb{E}\left[\rho_t\phi_t'\phi_t^\top\right].$$

The associated noise terms are given by

$$M_{t+1}^{(1)} = A_t\theta_t + b_t - C_t u_t - h^{(1)}(u_t, \theta_t, z_t),$$

$$M_{t+1}^{(2)} = A_t\theta_t + b_t - \gamma\rho_t\phi_t'\phi_t^\top u_t - h^{(2)}(u_t, \theta_t, z_t),$$

$$M_{t+1}^{(3)} = -\rho_t\delta_t\nabla_{z_t} \log \pi_{z_t}(a_t|s_t) - h^{(3)}(u_t, \theta_t, z_t).$$

By construction, $\{M_{t+1}^{(j)}\}_{t\geq 0}$, $j = 1, 2, 3$, are martingale difference sequences with respect to $\{\mathcal{F}_t\}_{t\geq 0}$. Under Assumptions 4–6, their conditional second moments satisfy linear-growth bounds.

The verification of the Lipschitz condition follows exactly as in Lakshminarayanan and Bhatnagar (2017), and we omit the details. Under the Gibbs parameterization, the policy-dependent coefficients are Lipschitz functions of the actor parameter, which in turn implies the required Lipschitz property of $h^{(1)}, h^{(2)}$, and $h^{(3)}$. Together with the martingale-difference and step-size conditions established above, this verifies **(A:1)**–**(A:3)**.

We now identify the limiting ODEs associated with the three timescales. For fixed $(\theta, z)$, the fastest-timescale ODE is

$$\dot{u}(t) = h^{(1)}(u(t), \theta, z) = \bar{A}_z\theta + \bar{b}_z - \bar{C}u(t).$$

Since $\bar{C}$ is positive definite, this ODE has the unique globally asymptotically stable equilibrium

$$\lambda^{(1)}(\theta, z) = \bar{C}^{-1}(\bar{A}_z\theta + \bar{b}_z).$$

Thus condition **(C.3.1)** is satisfied.

We next verify the corresponding scaled condition **(B.3.1)**. For $c \geq 1$, define

$$h_c^{(1)}(u, \theta, z) = \frac{h^{(1)}(cu, c\theta, cz)}{c} = \bar{A}_{cz}\theta + \frac{\bar{b}_{cz}}{c} - \bar{C}u.$$

As in Lakshminarayanan and Bhatnagar (2017), under the Gibbs parameterization the scaled policy $\pi_{cz}$ converges to the corresponding greedy policy $\pi_z^\infty$ uniformly on compacts as $c \to \infty$. Hence $\bar{A}_{cz} \to \bar{A}_z^\infty$ uniformly on compacts, while $\bar{b}_{cz}/c \to 0$. It follows that

$$h_c^{(1)}(u, \theta, z) \to h_\infty^{(1)}(u, \theta, z) = \bar{A}_z^\infty\theta - \bar{C}u$$

uniformly on compacts. Since $-\bar{C}$ is Hurwitz, the ODE

$$\dot{u}(t) = h_\infty^{(1)}(u(t), \theta, z)$$

has the unique globally asymptotically stable equilibrium

$$\lambda_\infty^{(1)}(\theta, z) = \bar{C}^{-1}\bar{A}_z^\infty\theta.$$

This establishes **(B.3.1)**.

We now turn to the intermediate timescale. Substituting the equilibrium $\lambda^{(1)}(\theta, z)$ into the second drift yields

$$
\begin{aligned}
\dot{\theta}(t) &= h^{(2)}(\lambda^{(1)}(\theta(t), z), \theta(t), z) \\
&= \bar{A}_z \theta(t) + \bar{b}_z - \gamma \bar{G}_z \bar{C}^{-1}(\bar{A}_z \theta(t) + \bar{b}_z).
\end{aligned}
$$

Equivalently,

$$
\dot{\theta}(t) = \left(\bar{A}_z - \gamma \bar{G}_z \bar{C}^{-1} \bar{A}_z\right)\theta(t) + \left(I - \gamma \bar{G}_z \bar{C}^{-1}\right)\bar{b}_z.
$$

Thus the reduced middle-timescale ODE is linear. Its equilibrium is given by

$$
\lambda^{(2)}(z) = -\left(\bar{A}_z - \gamma \bar{G}_z \bar{C}^{-1} \bar{A}_z\right)^{-1}\left(I - \gamma \bar{G}_z \bar{C}^{-1}\right)\bar{b}_z,
$$

provided the coefficient matrix is nonsingular. In the present setting, the verification of **(C.3.2)** follows similarly as in Lakshminarayanan and Bhatnagar (2017) and we therefore omit the details.

The scaled analogue **(B.3.2)** is obtained in the same manner. For $c \geq 1$, define

$$
h_c^{(2)}(\theta, z) = \frac{h^{(2)}(c\lambda_\infty^{(1)}(\theta, z), c\theta, cz)}{c}.
$$

Using the expression for $\lambda_\infty^{(1)}$ and again passing to the greedy-policy limit as in Lakshminarayanan and Bhatnagar (2017), we obtain

$$
h_c^{(2)}(\theta, z) \to h_\infty^{(2)}(\theta, z)
$$

uniformly on compacts, where $h_\infty^{(2)}(\theta, z)$ is the limiting reduced linear critic drift obtained after substituting the scaled faster-timescale equilibrium. Thus, the scaled middle-timescale condition reduces to the stability of the corresponding limiting linear critic ODE. We therefore conclude that the ODE

$$
\dot{\theta}(t) = h_\infty^{(2)}(\theta(t), z)
$$

has the unique globally asymptotically stable equilibrium $\lambda_\infty^{(2)}(z)$, thereby verifying **(B.3.2)**.

Now, define

$$
h_c^{(3)}(z) = \frac{h^{(3)}\left(c\lambda_\infty^{(1)}(\lambda_\infty^{(2)}(z), z), c\lambda_\infty^{(2)}(z), cz\right)}{c}.
$$

After passing to the scaled greedy-policy limit and using the fact that the critic-related terms are of lower order under the normalization by $c$, the limiting scaled actor drift reduces to

$$
h_\infty^{(3)}(z) = -\epsilon z.
$$

Consequently, the ODE

$$
\dot{z}(t) = h_\infty^{(3)}(z(t)) = -\epsilon z(t)
$$

has the origin as its unique globally asymptotically stable equilibrium, establishing **(B.3.3)**.

Finally, after substituting the faster-timescale equilibria into the actor drift, the slowest-timescale ODE becomes

$$
\dot{z}(t) = h^{(3)}\left(\lambda^{(1)}(\lambda^{(2)}(z(t)), z(t)), \lambda^{(2)}(z(t)), z(t)\right) = \hat{g}(z(t)) - \epsilon z(t),
$$

where

$$
\hat{g}(z) = -\mathbb{E}\left[\rho_t \delta_t(z) \nabla_z \log \pi_z(a_t|s_t)\right].
$$

This is precisely the regularized actor ODE. For sufficiently small $\epsilon > 0$, it may be viewed as a small perturbation of the unregularized actor ODE $\dot{z}(t) = \hat{g}(z(t))$; consequently, its trajectories converge to a small neighbourhood of the stationary set of the unregularized dynamics.

### 5.3 Constrained Actor-Critic

We now turn to the constrained average-cost setting. The notation is the same as in the previous subsection unless otherwise specified. In particular, $\phi(s) \equiv \phi_s \in \mathbb{R}^{d_1}$ denotes the critic feature associated with state $s$, and $\psi_{s,a} \in \mathbb{R}^{d_2}$ denotes the actor feature associated with the state-action pair $(s,a)$. We consider the problem

$$\min_z J(z) \qquad \text{subject to} \qquad G_k(z) \leq q_k, \quad 1 \leq k \leq N,$$

where

$$J(z) = \sum_s d^{\pi_z}(s) \sum_{a \in A(s)} \pi_z(a|s) \, c(s,a), \qquad G_k(z) = \sum_s d^{\pi_z}(s) \sum_{a \in A(s)} \pi_z(a|s) \, g_k(s,a),$$

and $d^{\pi_z}$ denotes the stationary distribution of the Markov chain induced by $\pi_z$. Introducing the multiplier vector

$$\bar{\gamma} = (\gamma_1, \ldots, \gamma_N)^\top \in \mathbb{R}_+^N,$$

the corresponding Lagrangian is

$$L(z, \bar{\gamma}) = J(z) + \sum_{k=1}^N \gamma_k \big(G_k(z) - q_k\big).$$

As in Bhatnagar and Lakshmanan (2012), we consider a three-timescale actor-critic scheme in which the critic estimates both the average Lagrangian cost and the corresponding differential value function, while the slowest timescale updates the multipliers. As in the previous subsection, we replace the projected actor update by an unprojected update regularized by an $\ell_2$ penalty. Thus, relative to Bhatnagar and Lakshmanan (2012), the only modification is the addition of the term $\epsilon z_n$ in the actor recursion.

We impose the following assumptions.

**Assumption 8.** *The one-step costs and features are uniformly bounded:*

$$|c(s,a)| \leq 1, \qquad |g_k(s,a)| \leq 1, \qquad \|\phi(s)\| \leq 1, \qquad \|\psi_{s,a}\| \leq 1$$

*for all $s \in \mathcal{S}$, $a \in \mathcal{A}$, and $1 \leq k \leq N$. Moreover, the critic feature matrix $\Phi$, whose $s$-th row is $\phi(s)^\top$, has full column rank.*

**Assumption 9.** *For each fixed actor parameter $z$, the Markov chain induced by $\pi_z$ admits a stationary distribution $d^{\pi_z}$, and the critic recursion is sampled in the corresponding stationary regime.*

**Assumption 10.** *The step-size sequences satisfy*

$$\alpha_n > 0, \quad \beta_n > 0, \quad \eta_n > 0, \qquad \sum_n \alpha_n = \sum_n \beta_n = \sum_n \eta_n = \infty,$$

$$\sum_n (\alpha_n^2 + \beta_n^2 + \eta_n^2) < \infty, \qquad \frac{\beta_n}{\alpha_n} \to 0, \qquad \frac{\eta_n}{\beta_n} \to 0.$$

*In addition, let $\tilde{\alpha}_n = K\alpha_n$ for some constant $K > 0$.*

Let

$$c_n = c(s_n, a_n), \qquad g_n = (g_{1,n}, \ldots, g_{N,n})^\top, \qquad \bar{q} = (q_1, \ldots, q_N)^\top.$$

The average-cost estimate $L_n$, critic parameter $v_n$, running constraint estimates $Y_n = (Y_1(n), \ldots, Y_N(n))^\top$, actor parameter $z_n$, and multiplier vector $\bar{\gamma}_n$ are updated according to

$$L_{n+1} = L_n + \tilde{\alpha}_n \Big(c_n + \bar{\gamma}_n^\top (g_n - \bar{q}) - L_n\Big), \tag{32}$$

$$\delta_n = c_n + \bar{\gamma}_n^\top (g_n - \bar{q}) - L_n + v_n^\top \big(\phi_{s_{n+1}} - \phi_{s_n}\big), \tag{33}$$

$$v_{n+1} = v_n + \alpha_n \, \delta_n \, \phi_{s_n}, \tag{34}$$

$$Y_{n+1} = Y_n + \alpha_n \big(g_n - Y_n\big), \tag{35}$$

$$z_{n+1} = z_n - \beta_n \big(\delta_n \, \psi_{s_n, a_n} + \epsilon z_n\big), \tag{36}$$

$$\bar{\gamma}_{n+1} = \Gamma \Big(\bar{\gamma}_n + \eta_n \big(Y_n - \bar{q}\big)\Big), \tag{37}$$

where $\Gamma$ denotes projection onto $\mathbb{R}_+^N$.

We now rewrite the recursion in the form required by the general three-timescale framework. Define the fast-timescale state

$$\zeta_n = \left(L_n,\ v_n^\top,\ Y_n^\top\right)^\top \in \mathbb{R}^{1+d_1+N}.$$

Then, for fixed $(z, \bar{\gamma})$, the fast recursion may be written in the affine form

$$\zeta_{n+1} = \zeta_n + \alpha_n\Big(A_n\zeta_n + B_n\bar{\gamma}_n + C_n\Big),$$

where

$$A_n = \begin{pmatrix} -K & 0_{1\times d_1} & 0_{1\times N} \\ -\phi_{s_n} & \phi_{s_n}(\phi_{s_{n+1}} - \phi_{s_n})^\top & 0_{d_1\times N} \\ 0_{N\times 1} & 0_{N\times d_1} & -I_N \end{pmatrix},$$

$$B_n = \begin{pmatrix} K(g_n - \bar{q})^\top \\ \phi_{s_n}(g_n - \bar{q})^\top \\ 0_{N\times N} \end{pmatrix}, \qquad C_n = \begin{pmatrix} Kc_n \\ c_n\phi_{s_n} \\ g_n \end{pmatrix}.$$

Similarly, the actor recursion can be written as

$$z_{n+1} = z_n + \beta_n\Big(-\epsilon z_n + X_n\zeta_n + Y_n^\psi\bar{\gamma}_n + Z_n\Big),$$

where

$$X_n = \left(-\psi_{s_n,a_n},\ \psi_{s_n,a_n}(\phi_{s_{n+1}} - \phi_{s_n})^\top,\ 0_{1\times N}\right), \qquad Y_n^\psi = \psi_{s_n,a_n}(g_n - \bar{q})^\top, \qquad Z_n = c_n\psi_{s_n,a_n}.$$

Therefore, the overall scheme takes the form

$$\zeta_{n+1} = \zeta_n + \alpha_n\big(h^{(1)}(\zeta_n, z_n, \bar{\gamma}_n) + M_{n+1}^{(1)}\big), \tag{38}$$

$$z_{n+1} = z_n + \beta_n\big(h^{(2)}(\zeta_n, z_n, \bar{\gamma}_n) + M_{n+1}^{(2)}\big), \tag{39}$$

$$\bar{\gamma}_{n+1} = \Gamma\Big(\bar{\gamma}_n + \eta_n\, h^{(3)}(\zeta_n, z_n, \bar{\gamma}_n)\Big), \tag{40}$$

for suitable drift functions $h^{(1)}, h^{(2)}, h^{(3)}$ and martingale-difference noise terms $M_{n+1}^{(1)}, M_{n+1}^{(2)}$.

We first verify **(A:1)**–**(A:3)**. The Lipschitz condition **(A:1)** follows exactly as in the previous subsection: $h^{(1)}$ is affine in $(\zeta, \bar{\gamma})$, with the dependence on $z$ entering only through policy-dependent coefficients, and the same reasoning applies to $h^{(2)}$ and $h^{(3)}$. For **(A:2)**, the boundedness of the one-step costs and features implies that the martingale-difference terms have conditional second moments with the required linear-growth bound. Finally, **(A:3)** follows directly from Assumption 10. We next identify the limiting ODEs associated with the three timescales. For fixed $(z, \bar{\gamma})$, the fast-timescale recursion tracks the linear ODE

$$\dot{\zeta}(t) = h^{(1)}(\zeta(t), z, \bar{\gamma}) = A^z\zeta(t) + B^z\bar{\gamma} + C^z,$$

where $A^z, B^z, C^z$ denote the corresponding stationary expectations under $\pi_z$. For each fixed $(z, \bar{\gamma})$, the fast-timescale dynamics reduce to the linear critic ODE

$$\dot{\zeta}(t) = A^z\zeta(t) + B^z\bar{\gamma} + C^z,$$

where $A^z$ is Hurwitz. Consequently, this ODE admits the unique globally asymptotically stable equilibrium

$$\lambda^{(1)}(z, \bar{\gamma}) = -(A^z)^{-1}(B^z\bar{\gamma} + C^z).$$

Thus **(C.3.1)** is satisfied. To verify **(B.3.1)**, define

$$h_{c,\bar{\gamma}}^{(1)}(\zeta, z) = \frac{h^{(1)}(c\zeta, cz, \bar{\gamma})}{c}.$$

As in the previous subsection, under the Gibbs parameterization the scaled policy $\pi_{cz}$ converges uniformly on compacts to the corresponding greedy policy $\pi_z^\infty$ as $c \to \infty$. Hence

$$h_{c,\bar{\gamma}}^{(1)}(\zeta, z) \to h_{\infty,\bar{\gamma}}^{(1)}(\zeta, z) := A^{\infty z}\zeta$$

uniformly on compacts. The limiting ODE

$$\dot{\zeta}(t) = A^{\infty z}\zeta(t)$$

has the origin as its unique globally asymptotically stable equilibrium. This establishes **(B.3.1)**.

Substituting the fast equilibrium into the actor recursion yields the reduced intermediate-timescale ODE

$$\dot{z}(t) = h^{(2)}(\lambda^{(1)}(z(t), \bar{\gamma}), z(t), \bar{\gamma}).$$

By construction of the critic and the policy-gradient estimator, this reduced drift corresponds to the regularized actor descent direction for the Lagrangian:

$$\dot{z}(t) = -\nabla_z L(z(t), \bar{\gamma}) - \epsilon z(t),$$

up to the sign convention corresponding to cost minimization. Thus the intermediate-timescale dynamics are given by the regularized actor ODE associated with the current multiplier. Accordingly, **(C.3.2)** holds provided this regularized actor ODE admits, for each fixed $\bar{\gamma}$, a unique globally asymptotically stable equilibrium.

For the associated scaled system, define

$$h_c^{(2)}(z, \bar{\gamma}) = \frac{h^{(2)}(c\lambda^{(1)}(z, \bar{\gamma}), cz, \bar{\gamma})}{c}.$$

As in the previous actor-critic example, the critic-dependent terms are lower order under the normalization by $c$, so that

$$h_c^{(2)}(z, \bar{\gamma}) \to -\epsilon z \qquad \text{uniformly on compacts.}$$

Therefore the limiting scaled ODE is

$$\dot{z}(t) = -\epsilon z(t),$$

whose unique globally asymptotically stable equilibrium is the origin. Thus **(B.3.2)** is satisfied.

Finally, after substituting both the fast and intermediate equilibria, the multiplier recursion tracks the ODE

$$\dot{\bar{\gamma}}(t) = G(z^{\bar{\gamma}(t)}) - \bar{q},$$

where

$$G(z) = \big(G_1(z), \ldots, G_N(z)\big)^\top$$

and $z^{\bar{\gamma}}$ denotes the attractor of the regularized actor ODE for fixed multiplier $\bar{\gamma}$. Equivalently,

$$\dot{\bar{\gamma}}(t) = \nabla_{\bar{\gamma}} L(z^{\bar{\gamma}(t)}, \bar{\gamma}(t)).$$

This is the typical dual-ascent interpretation of the multiplier dynamics for the constrained problem. In the spirit of Bhatnagar and Lakshmanan (2012), the slowest-timescale recursion may therefore be viewed as driving the iterates toward a constrained locally optimal solution.

## 6 Conclusions

In this work we have provided an easily verifiable set of sufficient conditions for stability and convergence of general $N$-timescale stochastic recursions with a martingale difference noise sequence, along with characterizing the limit of all the $N$ recursions. We then used these results to show that stochastic approximation algorithms with an added heavy ball term in the context of Gradient TD methods can be shown to converge

a.s. to the same TD solution asymptotically. There are several directions for further research. A natural direction to pursue, as mentioned in Remark 2, would be to come up with sufficient conditions for the stability and convergence of $N$-timescale stochastic recursions with Markov noise.

In Ramaswamy and Bhatnagar (2019), stability conditions for single timescale SA recursions with Markov noise have been provided, including the case where the underlying Markov process does not possess a unique stationary distribution and may depend on both the underlying parameters and an additional control sequence. More recently, Liu et al. (2025) extended the Borkar-Meyn stability theorem to the Markov noise setting in the single timescale case, under verifiable rate-of-change conditions on the underlying chain, with applications to off-policy reinforcement learning algorithms with eligibility traces. In the multi-timescale setting, Karmakar and Bhatnagar (2018) establishes convergence of two-timescale stochastic approximation with Markov noise assuming iterate-stability, while Ramaswamy and Bhatnagar (2016) analyses two-timescale recursions with set-valued maps in the absence of Markov noise, again assuming stability. Yaji and Bhatnagar (2020) subsumes both by analysing two-timescale stochastic approximation with set-valued maps and general Markov noise, once more under an iterate-stability assumption. Analysis of algorithms with set-valued maps is important as it paves the way for analysis of RL algorithms under partial observations/information. A natural extension of our results is thus towards deriving sufficient conditions for both stability and convergence of NN N-timescale stochastic approximation with (a) general Markov noise and (b) set-valued maps instead of the usual point-to-point maps.

Finally, Section 5 analyzed the asymptotic behaviour of the momentum algorithms. Further analysis of their finite time behaviour is called for to quantify the benefits of using momentum schemes in stochastic approximation. Towards this extension of weak convergence rate analysis of Konda and Tsitsiklis (2004); Mokkadem and Pelletier (2006) in the 2-TS setting and recent convergence rate results in expectation and high probability of 2-TS methods in Dalal et al. (2018b); Gupta et al. (2019); Kaledin et al. (2019); Dalal et al. (2020) to the $N$-timescale case would also be interesting directions to explore further.

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

# A  Proof of the Main Results

## A.1  Showing convergence by assuming Stability (Theorem 2)

We start by characterizing the set to which the iterate-vector $(x_n^{(1)}, x_n^{(2)}, \ldots, x_n^{(N)})$ converges. Extending the arguments in Lemma 1, Chapter 6 of Borkar (2008), we first consider the timescale of $\{\alpha_n^{(1)}\}$ and rewrite the $N$ iterates as follows:

$$x_{n+1}^{(1)} = x_n^{(1)} + \alpha_n^{(1)} \left( h^{(1)}\left(x_n^{(1)}, x_n^{(2)}, \ldots, x_n^{(N)}\right) + M_{n+1}^{(1)} \right),$$

$$x_{n+1}^{(2)} = x_n^{(2)} + \alpha_n^{(1)} \left( \epsilon_n^{(2)} + \bar{M}_{n+1}^{(2)} \right),$$

$$x_{n+1}^{(3)} = x_n^{(3)} + \alpha_n^{(1)} \left( \epsilon_n^{(3)} + \bar{M}_{n+1}^{(3)} \right),$$

$$\vdots$$

$$x_{n+1}^{(N)} = x_n^{(N)} + \alpha_n^{(1)} \left( \epsilon_n^{(N)} + \bar{M}_{n+1}^{(N)} \right),$$

where, we define $\forall 2 \leq j \leq N$, $\epsilon_n^{(j)} = \frac{\alpha_n^{(j)}}{\alpha_n^{(1)}} h^{(j)}\left(x_n^{(1)}, x_n^{(2)}, \ldots, x_n^{(N)}\right)$, $\quad \bar{M}_{n+1}^{(j)} = \frac{\alpha_n^{(j)}}{\alpha_n^{(1)}} M_{n+1}^{(j)}$.

From assumption **(A:3)**(iii), $\frac{\alpha_n^{(j)}}{\alpha_n^{(1)}} \to 0$, as $n \to \infty$, $\forall 2 \leq j \leq N$ and thus $\epsilon_n^{(j)} = o(1)$ a.s. Using now the third extension from Chapter-2 of Borkar (2008), $\left(x_n^{(1)}, x_n^{(2)}, \ldots, x_n^{(N)}\right)$ converges to an internally chain transitive invariant set of the ODE

$$\dot{x}^{(1)}(t) = h^{(1)}\left(x^{(1)}(t), x^{(2)}(t), \ldots, x^{(N)}(t)\right), \quad \dot{x}^{(2)}(t) = 0, \quad \dot{x}^{(3)}(t) = 0, \quad \ldots, \dot{x}^{(N)}(t) = 0.$$

For initial conditions $x^{(j)} \in \mathbb{R}^{d_j}, 1 \leq j \leq N$, the internally chain transitive invariant set of the above ODE system is $\left\{ \left( \lambda^{(1)}\left(x^{(2)}, x^{(3)}, \ldots, x^{(N)}\right), x^{(2)}, x^{(3)}, \ldots, x^{(N)}\right) \right\}$. Therefore,

$$\left(x_n^{(1)}, x_n^{(2)}, \ldots, x_n^{(N)}\right) \to \left\{ \left( \lambda^{(1)}\left(x^{(2)}, x^{(3)}, \ldots, x^{(N)}\right), x^{(2)}, x^{(3)}, \ldots, x^{(N)}\right), x^{(j)} \in \mathbb{R}^{d_j}, 2 \leq j \leq N \right\}. \quad (41)$$

Next, we consider the timescale $\alpha_n^{(m)}$, $2 \leq m \leq N-1$,

$$x_{n+1}^{(m)} = x_n^{(m)} + \alpha_n^{(m)}\left(h^{(m)}\left(x_n^{(1)}, x_n^{(2)}, \ldots, x_n^{(N)}\right) + M_{n+1}^{(m)}\right),$$

$$x_{n+1}^{(m+1)} = x_n^{(m+1)} + \alpha_n^{(m)}\left(\epsilon_n^{(m+1)} + \bar{M}_{n+1}^{(m+1)}\right),$$

$$\vdots$$

$$x_{n+1}^{(N)} = x_n^{(N)} + \alpha_n^{(m)}\left(\epsilon_n^{(N)} + \bar{M}_{n+1}^{(N)}\right),$$

where we re-define for $m+1 \leq j \leq N$ (note the abuse of notation)

$$\epsilon_n^{(j)} = \frac{\alpha_n^{(j)}}{\alpha_n^{(m)}} h^{(j)}\left(x_n^{(1)}, x_n^{(2)}, \ldots, x_n^{(N)}\right), \quad \bar{M}_{n+1}^{(j)} = \frac{\alpha_n^{(j)}}{\alpha_n^{(m)}} M_{n+1}^{(j)}.$$

From assumption **(A:3)**$(iii)$, $\frac{\alpha_n^{(j)}}{\alpha_n^{(m)}} \to 0$ as $n \to \infty$ and therefore $\epsilon_n^{(j)} = o(1)$ $\forall$ $m+1 \leq j \leq N$. Using the third extension from Chapter 2 of Borkar (2008), $\left(x_n^{(m+1)}, x_n^{(m+2)}, \ldots, x_n^{(N)}\right)$ converges to an internally chain transitive invariant set of the ODE

$$\dot{x}^{(m+1)}(t) = h^{(m+1)}\left(x^{(1)}(t), x^{(2)}(t), \ldots, x^{(N)}(t)\right), \quad \dot{x}^{(m+2)}(t) = 0, \quad \ldots, \quad \dot{x}^{(N)}(t) = 0.$$

For initial conditions $x^{(j)} \in \mathbb{R}^{d_j}$, $m \leq j \leq N$, the internally chain transitive invariant set of the ODE system is $\left\{\left(\lambda^{(m)}\left(x^{(m+1)}, \ldots, x^{(N)}\right), x^{(m+1)}, x^{(m+2)}, \ldots, x^{(N)}\right)\right\}$. Therefore,

$$\left(x_n^{(m)}, x_n^{(m+1)}, \ldots, x_n^{(N)}\right) \to \left\{\left(\lambda^{(m)}\left(x^{(m+1)}, x^{(m+2)}, \ldots, x^{(N)}\right), x^{(m+1)}, x^{(m+2)}, \ldots, x^{(N)}\right), x^{(j)} \in \mathbb{R}^{d_j}, m+1 \leq j \leq N\right\}.$$

We now infer the following from equation A.1:

$$\left(x_n^{(1)}, x_n^{(2)}, \ldots, x_n^{(N)}\right) \to \left\{\left(\lambda^{(1)}\left(\lambda^{(2)}\left(\ldots \lambda^{(N-2)}(\lambda^{(N-1)}(x^{(N)}), x^{(N)}), \ldots, x^{(N)}\right), x^{(N)}\right),\right.\right.$$

$$\left.\left.\lambda^{(2)}\left(\ldots \lambda^{(N-2)}\left(\lambda^{(N-1)}(x^{(N)}), x^{(N)}\right), \ldots, x^{(N)}\right), \ldots \ldots, \lambda^{(N)}(x^{(N)}), x^{(N)}\right) \,\middle|\, x^{(N)} \in \mathbb{R}^{d_N}\right\} \quad \text{a.s.} \quad (42)$$

Recall now that

$$\lambda^{(j:N-1)}(x) \triangleq \lambda^{(j)}\left(\lambda^{(j+1)}\left(\ldots \lambda^{(N-2)}(\lambda^{(N-1)}(x), x), \ldots, x\right), x\right), 1 \leq j \leq N-2.$$

Then, $\forall 1 \leq k \leq N-1$,

$$\left\|\lambda^{(k:N-1)}(x_n^{(N)}) - x_n^{(k)}\right\| \to 0 \text{ a.s. as } n \to \infty. \quad (43)$$

Finally, we consider the slowest timescale, i.e., the one corresponding to $\{\alpha_n^{(N)}\}$. We define the piece-wise linear continuous interpolation of the iterates $x_n^{(N)}$ as follows:

$$\bar{x}^{(N)}\left(t(n)\right) = x_n^{(N)},$$

$$\bar{x}^{(N)}(t) = x_n^{(N)} + \left(x_{n+1}^{(N)} - x_n^{(N)}\right)\frac{t - t(n)}{t(n+1) - t(n)}, \quad t \in [t(n), t(n+1)],$$

where, $t(n) = \sum_{m=0}^{n-1} \alpha_m^{(N)}, n \geq 1$ with $t(0) = 0$. Also, let $x^{(N),s}(t), t \geq s$, denote the unique solution of the below ODE starting at $s \in \mathbb{R}$:

$$\dot{x}^{(N),s}(t) = h^{(N)}\left(\lambda^{(1:N-1)}\left(x^{(N),s}(t)\right), \lambda^{(2:N-1)}\left(x^{(N),s}(t)\right), \ldots,\right.$$

$$\lambda^{(N-2:N-1)}\big(x^{(N),s}(t)\big), \lambda^{(N-1)}\big(x^{(N),s}(t)\big), x^{(n),s}(t)\bigg), \ t \geq s,$$

with $x^{(N),s}(s) = \bar{x}^{(N)}(s)$. Let $\psi_n^{(N)} = \sum_{i=0}^{n-1} \alpha_i^{(N)} M_{i+1}^{(N)}, \ n \geq 1$. It is easy to see that $\{\psi_n^{(N)}, \mathcal{F}_n\}$ is a zero-mean square integrable martingale. Further,

$$\sum_{n \geq 0} E[\| \ \psi_{n+1}^{(N)} - \psi_n^{(N)}\|^2 | \mathcal{F}_n] = \sum_{n \geq 0} (\alpha_n^{(N)})^2 E[\|M_{n+1}^{(N)}\|^2 | \mathcal{F}_n] < \infty,$$

a.s., from **(A:2)**, **(A:3)** and **(B.N.N+1)**. From the martingale convergence theorem, $\{\psi_n^{(N)}, \mathcal{F}_n\}$ is an almost surely convergent martingale sequence. Let $[t] = \max\{t(n) : t(n) \leq t\}, t \geq 0$. Then for $n, m \geq 0$.

$$\bar{x}^{(N)}(t(n+m)) = \bar{x}^{(N)}(t(n)) + \int_{t(n)}^{t(n+m)} h^{(N)}\bigg(\lambda^{(1:N-1)}\big(\bar{x}^{(N)}(t)\big), \lambda^{(2:N-1)}\big(\bar{x}^{(N)}(t)\big),$$

$$\ldots, \lambda^{(N-2:N-1)}\big(\bar{x}^{(N)}(t)\big), \lambda^{(N-1)}\big(\bar{x}^{(N)}(t)\big), \bar{x}^{(N)}(t)\bigg) + I + II + III,$$

where,

$$I = \left\| \int_{t(n)}^{t(n+m)} \bigg( h^{(N)}\Big(\lambda^{(1:N-1)}\big(\bar{x}^N([t])\big), \lambda^{(2:N-1)}\big(\bar{x}^N([t])\big), \ldots, \lambda^{(N-1)}\big(\bar{x}^N([t])\big), \bar{x}^N([t])\Big) \right.$$

$$\left. - h^{(N)}\Big(\lambda^{(1:N-1)}\big(\bar{x}^N(t)\big), \lambda^{(2:N-1)}\big(\bar{x}^N(t)\big), \ldots, \lambda^{(N-1)}\big(\bar{x}^N(t)\big), \bar{x}^N(t)\Big)\bigg) dt \right\|,$$

$$II = \left\| \sum_{j=0}^{m-1} \alpha_j^{(N)} \bigg( h^{(N)}\big(x_{n+j}^{(1)}, x_{n+j}^{(2)}, \ldots, x_{n+j}^{(N)}\big) \right.$$

$$\left. - h^{(N)}\big(\lambda^{(1:N-1)}\big(\bar{x}_{n+j}^N\big), \lambda^{(2:N-1)}\big(\bar{x}_{n+j}^N\big), \ldots, \lambda^{(N-1)}\big(\bar{x}_{n+j}^N\big), \bar{x}_{n+j}^N\big)\bigg) \right\|,$$

$$III = \left\| \big(\psi_{n+m+1}^{(N)} - \psi_n^{(N)}\big) \right\|.$$

Now, as in Lemma 2 of Chapter 2 in Borkar (2008), using Gronwall inequality we have:

$$\sup_{t \in [s,s+T]} \|\bar{x}^N(t) - x^{(N),s}(t)\| \leq K_T\big(I + II + III\big),$$

where, $K_T > 0$ is a constant that depends on $T$. We next show that all the three terms on the RHS above go to 0 as $n \to \infty$. Using Lipschitz continuity of $h^{(N)}$, we have

$$\|I\| \leq \int_{t(n)}^{t(n+m)} L\bigg( \left\|\lambda^{(1:N-1)}\big(\bar{x}^{(N)}([t])\big) - \lambda^{(1:N-1)}\big(\bar{x}^{(N)}(t)\big)\right\|$$

$$+ \left\|\lambda^{(2:N-1)}\big(\bar{x}^{(N)}([t])\big) - \lambda^{(2:N-1)}\big(\bar{x}^{(N)}(t)\big)\right\| + \cdots + \left\|\bar{x}^{(N)}([t]) - \bar{x}^{(N)}(t)\right\| \bigg)$$

$$\leq \sum_{j=0}^{m-1} K_L \int_{t(n+j)}^{t(n+j+1)} \left\|\bar{x}^{(N)}(t(n+j)) - \bar{x}^{(N)}(t)\right\|,$$

where $K_L = L(L^{N-1} + L^{N-2} + \cdots + L + 1)$ and we have used the fact that $\lambda^{(j:N)}$ is Lipschitz $\forall 1 \leq j \leq N-2$ and $[t] = t(n+j)$ for $t \in [t(n+j), t(n+j+1)], \forall \ 1 \leq j \leq m-1$. Now,

$$\|\bar{x}^{(N)}(t(n+j)) - \bar{x}^{(N)}(t)\| \leq \frac{\big(\|x_{n+j+1}^{(N)} - x_{n+j\|}^{(N)}\big) t}{t(n+j+1) - t(n+j)}.$$

Therefore, using A:3 (ii)

$$\|I\| \le \sum_{j=0}^{m-1} \mathcal{O}(\alpha_{n+j}^{(N)}) \int_{t(n+j)}^{t(n+j+1)} \frac{t}{t(n+j+1) - t(n+j)} dt \le \sum_{j>n} \mathcal{O}(\alpha_j^{(N)})^2 \to 0 \text{ as } n \to \infty.$$

Next, using Lipschitz continuity of $h^N(\cdot)$,

$$\|II\| \le \Big\| \sum_{j=0}^{m-1} \alpha_j^{(N)} L \sum_{k=1}^{N-1} \Big( \|x_{n+j}^{(k)} - \lambda^{(k:N-1)}(x_{n+j}^{(N)})\| \Big) \Big\|$$

$$\le L \sum_{k=1}^{N} \mathcal{O}\Big( \sup_{j \ge n} \Big\| x_j^{(k)} - \lambda^{(k:N-1)}(x_j^{(N)}) \Big\| \Big) \to 0 \text{ a.s. as } n \to \infty \text{ (from equation 43)}$$

Finally, from the foregoing, $\|III\| \le \sup_{j>n} \|\psi_j^{(N)} - \psi_n^{(N)}\| \to 0$ almost surely as $n \to \infty$. Now, using arguments as in Lemma 1 of Chapter 2 in Borkar (2008), it can be shown that

$$\lim_{s \to \infty} \sup_{t \in [s, s+T]} \|\bar{x}^{(N)}(t) - x^{(N),s}(t)\| = 0 \text{ a.s.}$$

Since, $x_*^{(N)}$ is a globally asymptotically stable equilibrium with $i = N$, using Theorem 2 of Chapter 2 in Borkar (2008), we get $x_n^{(N)} \to x_*^{(N)}$ a.s. as $n \to \infty$. Combining this with equation 42, proves Theorem 2.

## A.2 Showing Stability of the recursions (Theorem 3)

We begin with the fastest timescale recursion governed by the step-size sequence $\{\alpha_n^{(1)}\}$. We first state some of the notations and definitions used.

**(D1)** Let

$$t^{(1)}(n) = \sum_{i=0}^{n-1} \alpha_i^{(1)}, n \ge 1, \text{ with } t^{(1)}(0) = 0.$$

Let $X_k \triangleq (x_k^{(1)}, x_k^{(2)}, \ldots, x_k^{(N)})$, $k \ge 0$. For $t \in [t^{(1)}(n), t^{(1)}(n+1)]$,

$$\bar{X}^{(1)}(t) = X_n + (X_{n+1} - X_n) \frac{t - t^{(1)}(n)}{t^{(1)}(n+1) - t^{(1)}(n)}.$$

**(D2)** Given $t^{(1)}(n), n \ge 0$ and a constant $T > 0$ define

$$T_0 = 0, \text{ and } T_n = \min(t^{(1)}(m) : t^{(1)}(m) \ge T_{n-1} + T), n \ge 1.$$

One can find a subsequence $\{m(n)\}$ such that $T_n = t^{(1)}(m(n)) \ \forall n$ and $m(n) \to \infty$ as $n \to \infty$.

**(D3)** Define a sequence $r(n) \ge 1, \forall n$ as follows:

$$r(n) = \max(r(n-1), \|\bar{X}^{(1)}(T_n)\|, 1).$$

**(D4)** Define the scaled iterates (obtained from the recursion above) for $m(n) \le k \le m(n+1) - 1$ as:

$$\hat{x}_1^{(1)} \triangleq \frac{x_k^{(1)}}{r(n)}, \ \hat{x}_k^{(2)} \triangleq \frac{x_k^{(2)}}{r(n)}, \ldots, \hat{x}_k^{(N)} \triangleq \frac{x_k^{(N)}}{r(n)}.$$

Further,

$$\hat{x}_{k+1}^{(1)} = \hat{x}_k^{(1)} + \alpha_k^{(1)} \left( \frac{h^{(1)}(c\hat{x}_k^{(1)}, \ldots, c\hat{x}_k^{(N)})}{c} + \hat{M}_{k+1}^{(1)} \right),$$

$$\hat{x}_{k+1}^{(2)} = \hat{x}_k^{(2)} + \alpha_k^{(1)}\left(\epsilon_k^{(2)} + \hat{M}_{k+1}^{(2)}\right)$$

$$\vdots$$

$$\hat{x}_{k+1}^{(N)} = \hat{x}_k^{(N)} + \alpha_k^{(1)}\left(\epsilon_k^{(N)} + \hat{M}_{k+1}^{(3)}\right)$$

where, $c = r(n)$, and

$$\epsilon_k^{(j)} = \frac{\alpha_k^{(j)}}{\alpha_k^{(1)}}\frac{h^{(j)}(c\hat{x}_k^{(1)},\ldots,c\hat{x}_k^{(N)})}{c},\ 2 \le j \le N.$$

$$\hat{M}_{k+1}^{(1)} = \frac{M_{k+1}^{(1)}}{r(n)},\ \hat{M}_{k+1}^{(j)} = \frac{\alpha_k^{(j)}}{\alpha_k^{(1)}}\frac{M_{k+1}^{(j)}}{r(n)}, 2 \le j \le N.$$

**(D5)** Next we define the linearly interpolated trajectory for the scaled iterates $\hat{X}_n^{(1)} = (\hat{x}_n^{(1)}, \hat{x}_n^{(2)}, \ldots, \hat{x}_n^{(N)})$ as follows:

$$\hat{X}^{(1)}(t) = \hat{X}_n^{(1)} + (\hat{X}_{n+1}^{(1)} - \hat{X}_n^{(1)})\frac{t - t^{(1)}(n)}{t^{(1)}(n+1) - t^{(1)}(n)},\ \ t \in [t^{(1)}(n), t^{(1)}(n+1)].$$

**(D6)** Let $X_n^{(1)}(t) \triangleq (x_n^{(1)}(t), x_n^{(2)}(t), \ldots, x_n^{(N)}(t)), t \in [T_n, T_{n+1}]$ denote the trajectory of the ODE:

$$\dot{x}^{(1)}(t) = h_{r(n)}(x^{(1)}(t), x^{(2)}(t), \ldots, x^{(N)}(t)),\ \ \dot{x}^{(2)}(t) = 0,\ \ \ldots,\ \ \dot{x}^{(N)}(t) = 0,$$

with $x_n^{(j)}(T_n) = \hat{x}^{(j)}(T_n), \forall 1 \le j \le N$.

We state a lemma for an ODE with external inputs. Let $x_c^{(1),x^{(2)}(t),\ldots,x^{(N)}(t)}$ and $x_\infty^{(1),x^{(2)}(t),\ldots,x^{(N)}(t)}$ denote the trajectories of the following ODEs:

$$\dot{x}^{(1)}(t) = h_c^{(1)}\left(x^{(1)}(t), x^{(2)}(t), \ldots, x^{(N)}(t)\right),$$
$$\dot{x}^{(1)}(t) = h_\infty^{(1)}\left(x^{(1)}(t), x^{(2)}(t), \ldots, x^{(N)}(t)\right),$$

respectively, with initial condition $x^{(1)} \in \mathbb{R}^{d_1}$ and the external inputs $x^{(j)}(t) \in \mathbb{R}^{d_j}, \forall 2 \le j \le N$. Let $B^j(x^{(j)}, r) \triangleq \{q \in \mathbb{R}^{d_j} \big| \|q - x^{(j)}\| < r\}$ denote a ball of radius $r$ centred at $x^{(j)}$.

**Lemma 1.** *Let $x^{(1)} \in B^1(0,1) \subset \mathbb{R}^{d_1}, x^{(j)} \in W^{(j)} \subset \mathbb{R}^{d_j}, \forall 2 \le j \le N$ and let **(B.N.1)** hold. Then given $\epsilon > 0, \exists c_\epsilon \ge 1, r_\epsilon > 0$ and $T_\epsilon > 0$ such that for any external inputs satisfying $\tilde{x}^{(j)}(s) \in B^j(x^{(j)}, r_\epsilon)$, $\forall s \in [0, T], \forall c > c_\epsilon, t \ge T_\epsilon$,*

$$\|x_c^{(1),\tilde{x}^{(2)}(t),\ldots,\tilde{x}^{(N)}(s)}(t,x) - \lambda_\infty^{(1)}(x^{(2)}, \ldots, x^{(N)})\| \le 2\epsilon.$$

The next lemma uses the convergence result of $N$ scale iterates under the stability assumption of **(B.N.N+1)** and shows that the scaled iterates defined in **(D4)** converge.

**Lemma 2.** *Under **(A:1)-(A:3)**,*

(i) *For $0 \le k \le m(n+1) - m(n)$, $\|\hat{X}^{(1)}(t(m(n)+k))\| \le K^{(1)}$ a.s. for some constant $K^{(1)} > 0$.*

(ii) *$\lim_{n\to\infty} \sup_{t\in[T_n,T_{n+1}]} \|\hat{X}^{(1)}(t) - X_n^{(1)}(t)\| = 0$ a.s.*

The proof of the above two Lemmas follows in a similar manner as that of Lemmas 5 and 6, respectively, of Lakshminarayanan and Bhatnagar (2017). In particular, Lemma 2(i) shows that along the timescale of $\{\alpha_n^{(1)}\}$, between instants $T_n$ and $T_{n+1}$, the norm of the scaled iterate can grow at most by a factor $K^{(1)}$ starting from $B^1(0,1)$. Next, Lemma 2(ii) shows that the scaled iterate asymptotically tracks the ODE

defined in **(D6)**. The next lemma bounds $\|x_n^{(1)}\|$ in terms of $\|x_n^{(2)}\|, \ldots, \|x_n^{(N)}\|$. We define the linearly interpolated trajectories of the $N$ iterates as follows: $\bar{x}^{(j)}(t(n)) = x^{(j)}$ and $\forall t \in [t(n), t(n+1)]$,

$$\bar{x}^{(j)}(t) = x_n^{(j)} + \left(x_{n+1}^{(j)} - x_n^{(j)}\right) \frac{t - t^{(1)}(n)}{t^{(1)}(n+1) - t^{(1)}(n)}.$$

**Lemma 3.** *Under assumptions (A:1)-(A:3), $(B.N.i)_{1 \leq i \leq N}$ and $(C.N.i)_{1 \leq i \leq N}$,*

*(i) For $n$ large, and $T = T_{\frac{1}{4}}$, if $\|\bar{x}^{(1)}(T_n)\| > C_1(\|\bar{x}^{(2)}(T_n)\| + \cdots + \|\bar{x}^{(N)}(T_n)\|)$, for some $C_1 > 0$ then $\|\bar{x}^{(1)}(T_{n+1})\| \leq \frac{3}{4}\|\bar{x}^{(1)}(T_n)\|$.*

*(ii) $\|\bar{x}^{(1)}(T_n)\| \leq C_1^*(\|\bar{x}^{(2)}(T_n)\| + \cdots + \|\bar{x}^{(N)}(T_n)\|)$ a.s. for some $C_1^* \geq 1$.*

*(iii) $\|x_n^{(1)}\| \leq K_1^*\left(\|x_n^{(2)}\| + \cdots + \|x_n^{(N)}\|\right)$, for some $K_1^* > 0$.*

*Proof.* (i) We have $\|\bar{x}^{(1)}(T_n)\| > C_1(1 + \|\bar{x}^{(2)}(T_n)\| + \cdots + \|\bar{x}^{(N)}(T_n)\|)$. Since, $r(n) = \max(r(n-1), \|\bar{X}^{(1)}(T_n)\|, 1)$, $r(n) \geq \|\bar{X}^{(1)}(T_n)\|$. Therefore, $r(n) \geq C_1$. Next we show $\|\hat{x}^{(j)}(T_n)\| < \frac{1}{C_1}, \forall 2 \leq j \leq N$.

$$\text{For } p \geq 1, \quad \|\hat{x}^{(j)}(T_n)\|_p = \frac{\|\bar{x}^{(j)}(T_n)\|_p}{r(n)} \leq \frac{\|\bar{x}^{(j)}(T_n)\|_p}{\|\bar{X}^{(1)}(T_n)\|_p} = \frac{\|\bar{x}^{(j)}(T_n)\|_p}{\left(\|\bar{x}^{(1)}(T_n)\|_p^p + \cdots + \|\bar{x}^{(N)}(T_n)\|_p^p\right)^{\frac{1}{p}}}.$$

Since, $\|\bar{x}^{(1)}(T_n)\|_p \geq C_1(1 + \|\bar{x}^{(2)}(T_n)\|_p + \cdots + \|\bar{x}^{(N)}(T_n)\|_p)$,

$$\|\bar{x}^{(1)}(T_n)\|_p^p \geq C_1^p\left(\|\bar{x}^{(2)}(T_n)\|_p + \cdots + \|\bar{x}^{(N)}(T_n)\|_p\right)^p \geq C_1^p\left(\|\bar{x}^{(2)}(T_n)\|_p^p + \cdots + \|\bar{x}^{(N)}(T_n)\|_p^p\right),$$

$$\text{therefore,} \quad \|\hat{x}^{(2)}(T_n)\| \leq \frac{\|\bar{x}^{(2)}(T_n)\|_p}{\left(C_1^p + 1\right)^{\frac{1}{p}}\left(\|\bar{x}^{(2)}(T_n)\|_p^p + \cdots + \|\bar{x}^{(N)}(T_n)\|_p^p\right)^{\frac{1}{p}}} \leq \frac{1}{\left(1 + C_1^p\right)^{\frac{1}{p}}} < \frac{1}{C_1}.$$

The second inequality follows from the fact that $\|\bar{x}^{(2)}(T_n)\|_p^p \leq \|\bar{x}^{(2)}(T_n)\|_p^p + \cdots + \|\bar{x}^{(2)}(T_n)\|_p^p$. A similar analysis can be carried out to show that $\|\bar{x}^{(j)}(T_n)\|_p^p < \frac{1}{C_1}, 1 \leq j \leq N$. Next we show that $\|\hat{x}^{(1)}(T_n)\|_p > \frac{1}{1 + \frac{1}{C_1}}$. Here we focus on the case when iterates are blowing up. Therefore let $r(n) = \bar{X}^{(1)}(T_n)$. Then,

$$\|\hat{x}(T_n)\| = \frac{\|\bar{x}^{(1)}(T_n)\|}{\|\bar{X}^{(1)}(T_n)\|} = \frac{\|\bar{x}^{(1)}(T_n)\|}{\left(\|\bar{x}^{(1)}(T_n)\|_p^p + \cdots + \|\bar{x}^{(N)}(T_n)\|_p^p\right)^{\frac{1}{p}}}$$

$$= \frac{1}{\left(1 + \frac{\|\bar{x}^{(2)}(T_n)\|_p^p + \cdots + \|\bar{x}^{(N)}(T_n)\|_p^p}{\|\bar{x}^{(1)}(T_n)\|_p^p}\right)^{\frac{1}{p}}} > \frac{1}{\left(1 + \frac{\|\bar{x}^{(2)}(T_n)\|_p^p + \cdots + \|\bar{x}^{(N)}(T_n)\|_p^p}{C_1^p(\|\bar{x}^{(2)}(T_n)\|_p^p + \cdots + \|\bar{x}^{(N)}(T_n)\|_p^p)}\right)^{\frac{1}{p}}} > \frac{1}{1 + \frac{1}{C_1}}.$$

Let $\tilde{x}^{(j)}(t - T_n) = x_n^{(j)}(t), \forall 2 \leq j \leq N, \forall t \in [T_n, T_{n+1}]$. From Lemma 1, $\exists r_{\frac{1}{4}}, c_{\frac{1}{4}}, T_{\frac{1}{4}} > 0$ such that

$$\|x_c^{(1), \tilde{x}^{(2)}(t), \ldots, \tilde{x}^{(N)}(t)}(t, \hat{x}^{(1)}(T_n))\| \leq \frac{1}{4}, \forall t \geq T_{\frac{1}{4}}, \forall c \geq c_{\frac{1}{4}},$$

whenever $\tilde{x}^{(j)}(t) \in B^j(0, r_{\frac{1}{4}})$. Choose $C_1 > \max(c_{\frac{1}{4}}, \frac{2}{r_{\frac{1}{4}}})$ and $T = T_{\frac{1}{4}}$. Since $\dot{x}^{(j)}(t) = 0, \forall 2 \leq j \leq N$ for the ODE defined in **(D6)**, $\tilde{x}^{(j)}(t - T_n) = x_n^{(j)}(t) = \hat{x}^{(j)}(T_n), 2 \leq j \leq N, \forall t \in [T_n, T_{n+1}]$. From $\|\hat{x}^{(j)}(T_n)\| < \frac{1}{C_1}, 2 \leq j \leq N$, it follows that $\tilde{x}^{(j)}(s) \in B^j(0, r_{\frac{1}{4}}), 2 \leq j \leq N, \forall s \in [0, T]$. Using Lemma 2(ii), $\|\hat{x}^{(1)}(T_{n+1}^-) - x_n^{(1)}(T_{n+1})\| < \frac{1}{4}$ for large enough $n$. Also observe that $\|x_n^{(1)}(T_{n+1})\| = \|x_{r(n)}^{(1), \tilde{x}^{(2)}(t), \ldots, \tilde{x}^{(N)}(t)}(T_{n+1} - T_n, \hat{x}^{(1)}(T_n))\| \leq \frac{1}{4}$. Using these, we have $\|\hat{x}^{(1)}(T_{n+1}^-)\| \leq \|\hat{x}^{(1)}(T_{n+1}^-) - \bar{x}_n^{(1)}(T_{n+1})\| + \|\bar{x}_n^{(1)}(T_{n+1})\| \leq \frac{1}{2}$, where $\hat{x}^{(1)}(T_{n+1}^-) \triangleq \bar{x}^{(1)}(T_{n+1})/r(n)$. Finally, since

$$\frac{\|\bar{x}^{(1)}(T_{n+1})\|}{\|\bar{x}^{(1)}(T_n)\|} = \frac{\|\hat{x}^{(1)}(T_{n+1}^-)\|}{\|\hat{x}^{(1)}(T_n)\|},$$

we have, $\qquad \|\bar{x}^{(1)}(T_{n+1})\| = \dfrac{\|\hat{x}^{(1)}(T_{n+1}^-)\|}{\|\hat{x}^{(1)}(T_n)\|}\|\bar{x}^{(1)}(T_n)\| < \dfrac{\frac{1}{2}}{\frac{1}{1+1/C_1}}\|\bar{x}^{(1)}(T_n)\|.$

Choosing $C_1 > \max\left(c_{\frac{1}{4}}, \dfrac{2}{r_{\frac{1}{4}}}\right) > 2$, proves the claim.

The claims in (ii) and (iii) follow in a similar manner as Lemma 6(ii)-(iii) of Lakshminarayanan and Bhatnagar (2017). We repeat the arguments for the sake of completeness.

(ii) We will prove the claim by contradiction. Suppose there exists a monotonically increasing sequence $\{n_k\}$ such that $C_{n_k} \uparrow \infty$ as $k \to \infty$ and $\|\bar{x}^{(1)}(T_n)\| \geq C_{n_k}(\|\bar{x}^{(2)}(T_n)\| + \cdots + \|\bar{x}^{(N)}(T_n)\|)$, on a set of positive probability. From Lemma 2 (i), we know that if $\|\bar{x}^{(1)}(T_n)\| > C_1(\|\bar{x}^{(2)}(T_n)\| + \cdots + \|\bar{x}^{(N)}(T_n)\|)$, then $\|\bar{x}^{(1)}(T_k)\|$ falls at an exponential rate into a ball of radius $C_1(\|\bar{x}^{(2)}(T_k)\| + \cdots + \|\bar{x}^{(N)}(T_k)\|)$ for $k \geq n$. Therefore, corresponding to the sequence $\{n_k\}$, there exists another sequence $\{n_k'\}$ such that $\|\bar{x}^{(1)}(T_{n_{k-1}'})\| \leq C_1(\|\bar{x}^{(2)}(T_{n_k'})\| + \cdots + \|\bar{x}^{(N)}(T_{n_k'})\|)$ for $n_{k-1} \leq n_k' \leq n_k$, but $\|\bar{x}^{(1)}(T_{n_k'})\| > C_{n_k}(\|\bar{x}^{(2)}(T_{n_k'})\| + \cdots + \|\bar{x}^{(N)}(T_{n_k'})\|)$. From Lemma 2 (i), however, we know that the iterates $\bar{x}^{(1)}$ can only grow by a factor of $K^{(1)}$ between $m(n_k' - 1)$ and $m(n_k')$, leading to a contradiction. Therefore, $\|\bar{x}^{(1)}(T_n)\| \leq C_1^*(\|\bar{x}^{(2)}(T_n)\| + \cdots + \|\bar{x}^{(N)}(T_n)\|)$ a.s. for some $C_1^* \geq 1$.

(iii) From Lemma 2(ii), we know that $\forall t \in [T_n, T_{n+1}), \|\bar{x}^{(1)}(t)\| \leq K^{(1)}\|\bar{x}^{(1)}(T_n)\|$. Since, $\bar{x}^{(1)}(t)$ is a linear interpolation of the iterates $x_n^{(1)}$, therefore, $\|x_n^{(1)}\| \leq \sup_{t\in[t(n),t(n+1)]}\|\bar{x}^{(1)}(t)\|$. The claim therefore follows by choosing $K_1^* = C_1^* K^{(1)}$. $\qquad\square$

Next we consider the intermediate timescales of $\{\alpha_n^{(l)}\}_{n\geq 1}, 2 \leq l \leq N-1$, and re-define the terms below. Note the abuse of notation here when defining terms such as $T_n$, $r(n)$, $m(n)$, $\hat{M}_{n+1}^{(l)}$ etc., below.

**(E1)** Define

$$t^{(l)}(n) = \sum_{i=0}^{n-1} \alpha_i^{(l)}, n \geq 1 \text{ with } t^{(l)}(0) = 0.$$

Recall that $X_n = (x_n^{(1)}, x_n^{(2)}, \ldots, x_n^{(N)}), n \geq 0$. For $t \in [t^{(l)}(n), t^{(l)}(n+1)], l = 2, \ldots, N$, define

$$\bar{X}^{(l)}(t) = X_n + (X_{n+1} - X_n)\frac{t - t^{(l)}(n)}{t^{(l)}(n+1) - t^{(l)}(n)}.$$

**(E2)** Given $t^{(l)}(n), n \geq 0$ and a constant $T > 0$ define

$$T_0 = 0, \quad T_n = \min(t^{(l)}(m) : t^{(l)}(m) \geq T_{n-1} + T), n \geq 1$$

One can find a subsequence $\{m(n)\}$ such that $T_n = t^{(l)}(m(n)) \; \forall n$, and $m(n) \to \infty$ as $n \to \infty$.

**(E3)** The scaling sequence is defined as:

$$r(n) = \max(r(n-1)|\bar{X}^{(l)}(T_n)\|, 1), n \geq 1$$

**(E4)** The scaled iterates for $m(n) \leq k \leq m(n+1) - 1$ are defined by:

$$\hat{x}_k^{(1)} = \frac{x_k^{(1)}}{r(n)}, \hat{x}_k^{(2)} = \frac{x_k^{(2)}}{r(n)}, \ldots, \hat{x}_k^{(N)} = \frac{x_k^{(N)}}{r(n)}.$$

Further,

$$\hat{x}_{k+1}^{(1)} = \hat{x}_k^{(1)} + \alpha_k^{(1)}\left(\frac{h^{(1)}(c\hat{x}_k^{(1)}, \ldots, c\hat{x}_k^{(N)})}{c} + \hat{M}_{k+1}^{(1)}\right)$$

$$\hat{x}_{k+1}^{(2)} = \hat{x}_k^{(2)} + \alpha_k^{(2)}\left(\frac{h^{(2)}(c\hat{x}_k^{(1)}, \ldots, c\hat{x}_k^{(N)})}{c} + \hat{M}_{k+1}^{(2)}\right)$$

$$\vdots$$

$$\hat{x}_{k+1}^{(l)} = \hat{x}_k^{(l)} + \alpha_k^{(l)} \left( \frac{h^{(l)}(c\hat{x}_k^{(1)}, \ldots, c\hat{x}_k^{(N)})}{c} + \hat{M}_{k+1}^{(l)} \right)$$

$$\hat{x}_{k+1}^{(l+1)} = \hat{x}_k^{(l+1)} + \alpha_k^{(l)} \left( \epsilon_k^{(l+1)} + \hat{M}_{k+1}^{(l+1)} \right)$$

$$\vdots$$

$$\hat{x}_{k+1}^{(N)} = \hat{x}_k^{(N)} + \alpha_k^{(l)} \left( \epsilon_k^{(N)} + \hat{M}_{k+1}^{(N)} \right)$$

where, $c = r(n)$, and $\forall l \le j \le N - 1$,

$$\epsilon_k^{(j)} = \frac{\alpha_k^{(j)}}{\alpha_k^{(l)}} \frac{h^{(j)}(c\hat{x}_k^{(1)}, \ldots, c\hat{x}_k^{(N)})}{c},$$

$$\hat{M}_{k+1}^{(j)} = \frac{M_{k+1}^{(j)}}{r(n)}, \quad \text{for } 1 \le j \le l, \qquad \hat{M}_{k+1}^{(j)} = \frac{\alpha_k^{(j)}}{\alpha_k^{(l)}} \frac{M_{k+1}^{(j)}}{r(n)}, \quad \text{for } l+1 \le j \le N-1.$$

**(E5)** Next, we define the linearly interpolated trajectory for the scaled iterates $\hat{X}_n^{(l)} = (x_n^{(1)}, x_n^{(2)}, \ldots, x_n^{(N)})$ for $t \in [t^{(l)}(n), t^{(l)}(n+1)]$ as follows:

$$\hat{X}^{(l)}(t) = \hat{X}_n^{(l)} + (\hat{X}_{n+1}^{(l)} - \hat{X}_n^{(l)}) \frac{t - t(n)}{t(n+1) - t(n)}.$$

**(E6)** Let $X_n^{(l)}(t) = (x_n^{(1)}(t), \ldots, x_n^{(N)}(t)), t \in [T_n, T_{n+1}]$ denote the trajectory of the ODE:

$$\dot{x}^{(1)}(t) = h_{r(n)}^{(1)}(x^{(1)}(t), \ldots, x^{(N)}(t)),$$

$$\dot{x}^{(2)}(t) = h_{r(n)}^{(2)}(x^{(2)}(t), \ldots, x^{(N)}(t)),$$

$$\vdots$$

$$\dot{x}^{(l)}(t) = h_{r(n)}^{(l)}(x^{(l)}(t), \ldots, x^{(N)}(t)),$$

$$\dot{x}^{(l+1)}(t) = 0$$

$$\vdots$$

$$\dot{x}^{(N)}(t) = 0$$

with $x_n^{(j)}(T_n) = \hat{x}^{(j)}(T_n), \forall 1 \le j \le N$. We refer the reader to Assumption **(B.N.i)** for the definition of $h_{r(n)}^{(i)}(x^{(i)}(t), \ldots, x^{(N)}(t)), i = 1, \ldots, N$.

**Lemma 4.** *Let $x^{(l)} \in B^l(0, 1) \subset \mathbb{R}^{d_l}, x^{(j)} \in W^{(j)} \subset \mathbb{R}^{d_j}, l+1 \le j \le N$ and let **(B.N.l)** hold. Then given $\epsilon > 0, \exists c_\epsilon \ge 1, r_\epsilon > 0$ and $T_\epsilon > 0$ such that for any external inputs satisfying $\tilde{x}^{(j)}(s) \in B^j(x^{(j)}, r_\epsilon), l+1 \le j \le N-1, \forall s \in [0, T], \forall c > c_\epsilon, t \ge T_\epsilon,$*

$$\|x_c^{(l), \tilde{x}^{(l+1)}(t), \ldots, \tilde{x}^{(N)}(s)}(t, x^{(l)}) - \lambda_\infty^{(l)}(x^{(l+1)}, \ldots, x^{(N)})\| \le 2\epsilon,$$

*with $x_c^{(l), \tilde{x}^{(l+1)}(t), \ldots, \tilde{x}^{(N)}(s)}(t, x^{(l)})$ defined analogously as in Lemma 1.*

**Lemma 5.** *Under **(A:1)-(A:3)**,*

(i) *For $0 \leq k \leq m(n+1) - m(n)$, and $2 \leq l \leq N - 1$ $\|\hat{X}^{(l)}(t(m(n)+k))\| \leq K^{(l)}$ a.s. for some constant $K^{(l)} > 0$.*

(ii) *We have, $\lim_{n\to\infty} \sup_{t\in[T_n, T_{n+1}]} \|\hat{X}^{(l)}(t) - X_n^{(l)}(t)\| = 0$ a.s.*

The proof of the above two Lemmas follows in a similar manner as the proof of Lemmas 5 and 9, respectively, of Lakshminarayanan and Bhatnagar (2017).

**Lemma 6.** *Assume (A:1)-(A:3), (B.N.i)$_{1\leq i\leq N}$ and (C.N.i)$_{1\leq i\leq N}$ hold. Then,*

(i) *For $n$ large enough, there exists $T > 0$, such that if $\|\bar{x}^{(l)}(T_n)\| > C_l(\|\bar{x}^{(l+1)}(T_n)\| + \cdots + \|\bar{x}^{(N)}(T_n)\|)$, for some $C_l > 0$, then $\|\bar{x}^{(l)}(T_{n+1})\| < \frac{3}{4}\|\bar{x}^{(l)}(T_n)\|$.*

(ii) *$\|\bar{x}^{(l)}(T_n)\| \leq C_l^* \left( \|\bar{x}^{(l+1)}(T_n)\| + \cdots + \|\bar{x}^{(N)}(T_n)\| \right)$, for some $C_l^* > 1$.*

(iii) *$\|x_n^{(l)}\| \leq K_l^*(\|x_n^{(l+1)}\| + \cdots + \|x_n^{(N)}\|)$, for some $K_l^* > 0$.*

*Proof.* We use an inductive argument to show that the lemma holds. The base case $l = 1$ holds from Lemma 3. Assume the claim holds for $l - 1$. We now show that it holds for $l$. As in Lemma 3, we can show $\|\bar{x}^{(j)}(T_n)\|_p^p < \frac{1}{C_l}, l + 1 \leq j \leq N$. Next we show that $\|\hat{x}^{(l)}(T_n)\| > \frac{1}{((l-1)(C_{l,max}^*)^{l-1}+1)(1+\frac{1}{C_l})}$, where $C_{l,max}^* = \max_{1\leq j\leq l-1}(C_j^*)$. Here again we are considering the case when the iterates are blowing up. Therefore let $r(n) = \|\bar{X}^{(l)}(T_n)\|$. Now, from the inductive step, we know that $\|\bar{x}^{(j)}(T_n)\| \leq C_j^*(\|\bar{x}^{(j+1)}(T_n)\| + \cdots + \|\bar{x}^{(N)}(T_n)\|), l \leq j \leq N$ and therefore,

$$r(n) \leq \sum_{j=1}^{l-1} C_j^*(\|\bar{x}^{(j+1)}(T_n)\| + \cdots + \|\bar{x}^{(N)}(T_n)\|) + \|\bar{x}^{(l)}(T_n)\| + \cdots + \|\bar{x}^{(N)}(T_n)\|$$

$$= \Big(\sum_{j=1}^{l-1}\prod_{v=1}^{l-j} C_v^* + 1\Big)(\|\bar{x}^{(l)}(T_n)\| + \cdots + \|\bar{x}^{(N)}(T_n)\|) \leq \Big((l-1)(C_{l,max}^*)^{(l-1)} + 1\Big)\Big(\|\bar{x}^{(l)}(T_n)\| + \cdots + \|\bar{x}^{(N)}(T_n)\|\Big).$$

We thus have,

$$\|\hat{x}^{(l)}(T_n)\|_p \geq \frac{\|\bar{x}^{(l)}(T_n)\|}{((l-1)(C_{l,max}^*)^{l-1}+1)(\|\bar{x}^{(l)}(T_n)\| + \cdots + \|\bar{x}^{(N)}(T_n)\|)}$$

$$> \frac{1}{((l-1)(C_{l,max}^*)^{l-1}+1)(1+\frac{1}{C_l})} = \frac{\epsilon}{1+\frac{1}{C_l}},$$

where $\epsilon = \frac{1}{(l-1)(C_{l,max}^*)^{l-1}+1}$. Let $\tilde{x}^{(j)}(t - T_n) = x_n^{(j)}(t), \forall 2 \leq j \leq N, \forall t \in [T_n, T_{n+1}]$. From Lemma 4, $\exists r_{\epsilon/4}, c_{\epsilon/4}, T_{\epsilon/4} > 0$ such that

$$\|x_c^{(l),\tilde{x}^{(l+1)}(t),\dots,\tilde{x}^{(N)}(t)}(t, \hat{x}^{(1)}(T_n))\| \leq \epsilon,$$

$\forall t \geq T_{\epsilon/4}, \forall c \geq c_{\epsilon/4}$, whenever $\tilde{x}^{(j)}(t) \in B^j(0, r_{\epsilon/4}), l + 1 \leq j \leq N$. Choose $T = T_{\epsilon/4}$. Since $\dot{x}^{(j)}(t) = 0$, $l + 1 \leq j \leq N$ for the ODE defined in **(E6)** and $\tilde{x}^{(j)}(t - T_n) = x_n^{(j)}(t) = \hat{x}^{(j)}(T_n) \forall t \in [T_n, T_{n+1}]$ and we choose $C_l > \max\left(c_{\epsilon/4}, \frac{2}{r_{\epsilon/4}}\right)$, from $\|\hat{x}^{(j)}(T_n)\| < \frac{1}{C_l}$, it follows that $\tilde{x}^{(j)}(s) \in B^j(0, r_{\epsilon/4}), \forall s \in [0, T]$. Using Lemma 5(ii), $\exists \Gamma_1 > 0$ s.t. $\|\hat{x}^{(l)}(T_{n+1}^-) - x_n^{(l)}(T_{n+1})\| < \frac{\epsilon}{4}$ for large enough $n$ and $r(n) > \Gamma_l$. Choose $C_l > \max\left(c_{\epsilon/4}, \frac{2}{r_{\epsilon/4}}, C_1\right)$. Also observe that

$$\|x_n^{(l)}(T_{n+1})\| = \|x_{r(n)}^{(l),\tilde{x}^{(l+1)}(t),\dots,\tilde{x}^{(N)}(t)}(T_{n+1} - T_n, \hat{x}^{(l)}(T_n))\| \leq \frac{\epsilon}{4}.$$

Using these, we have

$$\|\hat{x}^{(l)}(T_{n+1}^-)\| \le \|\hat{x}^{(l)}(T_{n+1}^-) - x_n^{(l)}(T_{n+1})\| + \|x_n^{(l)}(T_{n+1})\| \le \frac{\epsilon}{2}.$$

Finally, since

$$\frac{\|\bar{x}^{(l)}(T_{n+1})\|}{\|\bar{x}^{(l)}(T_n)\|} = \frac{\|\hat{x}^{(l)}(T_{n+1}^-)\|}{\|\hat{x}^{(l)}(T_n)\|},$$

we have

$$\|\bar{x}^{(l)}(T_{n+1})\| = \frac{\|\hat{x}^{(l)}(T_{n+1}^-)\|}{\|\hat{x}^{(l)}(T_n)\|}\|\bar{x}^{(l)}(T_n)\| < \frac{\frac{\epsilon}{2}}{\frac{\epsilon}{((l-1)((1+1/C_l))}}\|\bar{x}(T_n)\| < \frac{1 + \frac{1}{C_l}}{2}$$

Choosing $C_l > 2$ proves the claim.

(ii) As before, we will prove the claim by contradiction. Suppose there exists a monotonically increasing sequence $\{n_k\}$ such that $C_{n_k} \uparrow \infty$ as $k \to \infty$ and $\|\bar{x}^{(l)}(T_n)\| \ge C_{n_k}(\|\bar{x}^{(l+1)}(T_n)\| + \cdots + \|\bar{x}^{(N)}(T_n)\|)$, on a set of positive probability. From Lemma 5 (i), we know that if $\|\bar{x}^{(l)}(T_n)\| > C_l(\|\bar{x}^{(l+1)}(T_n)\| + \cdots + \|\bar{x}^{(N)}(T_n)\|)$, then $\|\bar{x}^{(l)}(T_k)\|$ falls at an exponential rate into a ball of radius $C_l(\|\bar{x}^{(l+1)}(T_k)\| + \cdots + \|\bar{x}^{(N)}(T_k)\|)$ for $k \ge n$. Therefore, corresponding to the sequence $\{n_k\}$, there exists another sequence $\{n_k'\}$ such that $\|\bar{x}^{(l)}(T_{n_{k-1}'})\| \le C_l(\|\bar{x}^{(l+1)}(T_{n_k'})\| + \cdots + \|\bar{x}^{(N)}(T_{n_k'})\|)$ for $n_{k-1} \le n_k' \le n_k$, but $\|\bar{x}^{(l)}(T_{n_k'})\| > C_{n_k}(\|\bar{x}^{(l+1)}(T_{n_k'})\| + \cdots + \|\bar{x}^{(N)}(T_{n_k'})\|)$. From Lemma 5 (i), however, we know that the iterates $\bar{x}^{(l)}$ can only grow by a factor of $K^{(l)}$ between $m(n_k' - 1)$ and $m(n_k')$, leading to a contradiction. Therefore, $\|\bar{x}^{(l)}(T_n)\| \le C_l^*(\|\bar{x}^{(l+1)}(T_n)\| + \cdots + \|\bar{x}^{(N)}(T_n)\|)$ a.s. for some $C_l^* \ge 1$.

(iii) From Lemma 2(ii), we know that $\forall t \in [T_n, T_{n+1})$, $\|\bar{x}^{(l)}(t)\| \le K^{(l)}\|\bar{x}^{(l)}(T_n)\|$. Since, $\bar{x}^{(l)}(t)$ is a linear interpolation of the iterates $x_n^{(l)}$, therefore, $\|x_n^{(l)}\| \le \sup_{t \in [t(n),t(n+1)]} \|\bar{x}^{(l)}(t)\|$. The claim therefore follows by choosing $K_l^* = C_l^* K^{(l)}$. $\qquad\square$

Finally, we consider the slowest timescale recursion corresponding to the step-size $\{\alpha_n^{(N)}\}_{n \ge 0}$, and re-define the terms used. Note (again) the abuse of notation in definitions of terms such as $T_n$, $m(n)$, $r(n)$, $\hat{M}_{n+1}^j$, etc., below.

**(F1)** Define $t^{(N)}(n) = \sum_{i=0}^{n-1} \alpha_i^{(N)}$, $n \ge 1$ with $t(0) = 0$. Let $X_k^{(N)} = (x_k^{(1)}, x_k^{(2)}, \ldots, x_k^{(N)})$, $k \ge 0$, and for $t \in [t^{(N)}(n), t^{(N)}(n+1)]$,

$$\bar{X}^{(N)}(t) = X_n^{(N)} + \left(X_{n+1}^{(N)} - X_n^{(N)}\right)\frac{t - t^{(N)}(n)}{t^{(N)}(n+1) - t^{(N)}(n)}.$$

**(F2)** Given $t^{(N)}(n)$, $n \ge 0$ and constant $T > 0$, define

$$T_n = \min(t^{(N)}(m) : t^{(N)}(m) \ge T_{n-1} + T), n \ge 1,$$

with $T_0 = 0$. One can find a subsequence $\{m(n)\}$ such that $T_n = t^{(N)}(m(n))$, $\forall n$, and $m(n) \to \infty$ as $n \to \infty$.

**(F3)** The scaling sequence is defined as $r(n) = \max(r(n-1), \|\bar{X}^{(N)}(T_n)\|, 1)$, $n \ge 1$.

**(F4)** The scaled iterates for $m(n) \le k \le m(n+1) - 1$ are given by

$$\hat{x}_k^{(1)} = \frac{x_k^{(1)}}{r(n)}, \hat{x}_k^{(2)} = \frac{x_k^{(2)}}{r(n)}, \ldots, \hat{x}_k^{(N)} = \frac{x_k^{(N)}}{r(n)}.$$

The corresponding updates are as follows:

$$\hat{x}_{k+1}^{(1)} = \hat{x}_k^{(1)} + \alpha_k^{(1)}\left(\frac{h^{(1)}(c\hat{x}_k^{(1)}, \ldots, c\hat{x}_k^{(N)})}{c} + \hat{M}_{k+1}^{(1)}\right),$$

$$\vdots$$

$$\hat{x}_{k+1}^{(N)} = \hat{x}_k^{(N)} + \alpha_k^{(N)} \left( \frac{h^{(N)}(c\hat{x}_k^{(1)}, \ldots, c\hat{x}_k^{(N)})}{c} + \hat{M}_{k+1}^{(N)} \right),$$

where, $c = r(n)$, and $\forall 1 \leq j \leq N - 1$, $\hat{M}_{k+1}^{(j)} = \dfrac{M_{k+1}^{(j)}}{r(n)}$, for $1 \leq j \leq N$.

**(F5)** Next, we define the linearly interpolated trajectory for the scaled iterates $\hat{X}_n^{(N)} = (x_n^{(1)}, x_n^{(2)}, \ldots, x_n^{(N)})$ as follows: For $t \in [t^{(N)}(n), t^{(N)}(n+1)]$,

$$\hat{X}^{(N)}(t) = \hat{X}_n^{(N)} + (\hat{X}_{n+1}^{(N)} - \hat{X}_n^{(N)}) \frac{t - t^N(n)}{t^N(n+1) - t^N(n)}.$$

**(F6)** Let $X_n^{(N)}(t) = (x_n^{(1)}(t), \ldots, x_n^{(N)}(t)), t \in [T_n, T_{n+1}]$, denote the trajectory of the ODE:

$$\dot{x}^{(1)}(t) = h_{r(n)}^{(1)}(x^{(1)}(t), \ldots, x^{(N)}(t)),$$

$$\dot{x}^{(2)}(t) = h_{r(n)}^{(2)}(x^{(2)}(t), \ldots, x^{(N)}(t)),$$

$$\vdots$$

$$\dot{x}^{(N)}(t) = h_{r(n)}^{(N)}(x^{(N)}(t)),$$

with $x_n^{(j)}(T_n) = \hat{x}^{(j)}(T_n)$, $1 \leq j \leq N$.

**Lemma 7.** *Let $x^{(N)} \in B^N(0,1) \subset \mathbb{R}^{d_N}$ and let Assumption (**B.N.N**) hold. Then given $\epsilon > 0, \exists c_\epsilon \geq 1, r_\epsilon > 0$ and $T_\epsilon > 0$, then*

$$\|x_c^{(N)}(t, x^{(N)})\| \leq 2\epsilon, \quad \forall c > c_\epsilon.$$

**Lemma 8.** *Under (A:1)-(A:3),*

(i) *For $0 \leq k \leq m(n+1) - m(n)$, $\|\hat{X}^{(N)}(t(m(n)+k))\| \leq K^{(N)}$ a.s. for some constant $K^{(N)} > 0$.*

(ii) *For sufficiently large $n$, we have $\sup_{[T_n, T_{n+1})} \|\hat{x}^{(N)}(t) - x_n^{(N)}(t)\| \to 0$ almost surely as $c \to \infty$.*

As before, the proof of the above two Lemmas follows in a similar manner as the proof of Lemmas 5 and 9, respectively, of Lakshminarayanan and Bhatnagar (2017).

**Lemma 9.** *Under assumptions (A:1)-(A:3), (B.N.i)$_{1 \leq i \leq N}$ and (C.N.i)$_{1 \leq i \leq N}$, we have:*

(i) *For $n$ large, $\exists T$ such that if $\|\bar{x}^{(N)}(T_n)\| > C_N$, for some $C_N > 0$ then $\|\bar{x}^{(N)}(T_{n+1})\| < \frac{1}{2}\|\bar{x}^{(N)}(T_n)\|$.*

(ii) *$\|\bar{x}^{(N)}(T_n)\| \leq C_N^*$ for some $C_N^* > 0$.*

(iii) *$\sup_n \|x_n^{(N)}\| < \infty$ a.s.*

*Proof.* (i) From Lemmas 3 and 6, we know that

$$r(n) \leq \sum_{j=1}^{N-1} C_j^* (\|\bar{x}^{(j+1)}(T_n)\| + \cdots + \|\bar{x}^{(N)}(T_n)\|) = \left( \sum_{j=1}^{N-1} \prod_{v=1}^{N-j} C_v^* + 1 \right) \left( \|\bar{x}^{(N)}(T_n)\| \right).$$

Therefore, $\|\hat{x}^{(N)}(T_n)\| = \dfrac{\|\bar{x}^{(N)}(T_n)\|}{r(n)} > \dfrac{1}{\sum_{j=1}^{N-1} \left( \prod_{v=1}^{N-j} C_v^* + 1 \right)} > \dfrac{1}{((N-1)(C_{N,max}^*)^{N-1} + 1)} \triangleq \tilde{\epsilon},$

where, $C_{N,max}^* = \max\limits_{1 \le j \le N-1}(C_j^*)$. Since, $0 \in \mathbb{R}^{d_3}$ is the unique globally asymptotically stable equilibrium, using Lemma 7, $\exists c_{\tilde{\epsilon}/4}, T_{\tilde{\epsilon}/4} > 0$, such that $\|x_c^{(N)}(t, x^{(N)})\| < \frac{\tilde{\epsilon}}{4}$, $\forall c \ge c_{\tilde{\epsilon}/4}, t \ge T_{\tilde{\epsilon}/4}$. Also, for sufficiently large $n$, from Lemma 8(ii), $\exists \Gamma_2 > 0$ such that $\|\hat{x}^{(N)}(T_{n+1}^-) - x_n^{(N)}(T_{n+1})\| < \frac{\tilde{\epsilon}}{4}$ for $r(n) > \Gamma_2$. We pick $C_N = \max(c_{\tilde{\epsilon}/4}, \Gamma_2)$ and $T = T_{\tilde{\epsilon}/4}$. For $n$ large, it follows that

$$\|\hat{x}^{(N)}(T_{n+1}^-)\| \le \|\hat{x}^{(N)}(T_{n+1}^-) - x_n^{(N}(T_{n+1})\| + \|x_n^{(N)}(T_{n+1})\| \le \frac{\tilde{\epsilon}}{4}.$$

Finally, since $\dfrac{\|\bar{x}^{(N)}(T_{n+1})\|}{\|\bar{x}^{(N)}(T_n)\|} = \dfrac{\|\hat{x}^{(N)}(T_{n+1}^-)\|}{\|\hat{x}^{(N)}(T_n)\|}$, it follows that $\|\bar{x}^{(N)}(T_{n+1})\| < \frac{1}{2}\|\bar{x}^{(N)}(T_n)\|$.

The claims in (ii) and (iii) now follow in a similar manner as Theorem 10 (iii)-(iv), respectively, of Lakshminarayanan and Bhatnagar (2017), and the arguments are repeated here for completeness.

(ii) We will prove the claim by contradiction. Suppose there exists a monotonically increasing sequence $\{n_k\}$ such that $C_{n_k} \uparrow \infty$ as $k \to \infty$ and $\|\bar{x}^{(N)}(T_n)\| \ge C_{n_k}$, on a set of positive probability. From Lemma 8 (i), we know that if $\|\bar{x}^{(N)}(T_n)\| > C_N$, then $\|\bar{x}^{(N)}(T_k)\|$ falls at an exponential rate into a ball of radius $C_N$ for $k \ge n$. Therefore, corresponding to the sequence $\{n_k\}$, there exists another sequence $\{n'_k\}$ such that $\|\bar{x}^{(N)}(T_{n'_{k-1}})\| \le C_N$ for $n_{k-1} \le n'_k \le n_k$, but $\|\bar{x}^{(N)}(T_{n'_k})\| > C_{n_k}$. From Lemma 8 (i), however, we know that the iterates $\bar{x}^{(N)}$ can only grow by a factor of $K^{(N)}$ between $m(n'_k - 1)$ and $m(n'_k)$, leading to a contradiction. Therefore, $\|\bar{x}^{(N)}(T_n)\| \le C_N^*$ a.s. for some $C_N^* \ge 1$.

(ii) From the previous part we have $\|\bar{x}^{(N)}\| \le C_N^*$, and from Lemma 8 (i), we have $\|\bar{x}^{(N)}(t)\| \le K^{(N)}\|\bar{x}^{(N)}(T_n)\|$, $\forall t \in [T_n, T_{n+1})$. Therefore, $\|x_n\| \le K^{(N)}C_N^*$ almost surely. $\qquad \square$

The fact that $\sup_n \|x_n^{(l)}\| < \infty$ a.s. $\forall l \in \{1, \dots, N\}$ follows by combining Lemma 9 (iii) and Lemma 6 (iii).

