# OpenReview forum: "Multi Timescale Stochastic Approximation: Stability and Convergence"
_TMLR — Under review for TMLR_

### Review · Reviewer_idpo · 2026-07-01

**Summary Of Contributions:**

This paper claims to establish the stability and convergence of $N$-timescale stochastic approximation under i.i.d. noise. It largely relies on the methods of Lakshminarayanan and Bhatnagar (2017) in the two-timescale case and inductively extends them to an arbitrary number of timescales $N$. It then presents several applications in reinforcement learning and provides experiments demonstrating implementations of some representative algorithms where this $N$-timescale framework applies.

Strengths:
The main text is detailed and well-written.
There are several algorithms where the $N$-timescale framework is necessary, and the experiments help motivate their utility.

Weaknesses:
While the main text is generally well-presented, there is some sloppiness, especially in the proofs, with respect to the mathematics. Additionally, there are technical issues with (at least) the stability proof.
While the applications are of interest, the proofs of the main stochastic approximation results seem uninteresting, as they are a rather straightforward extension of the methods of Lakshminarayanan and Bhatnagar (2017).

**Audience:**

Yes

**Audience Explanation:**

This is a soft yes. Those interested in stochastic approximation would likely be interested in the paper’s findings. However, I do have some points for the authors to address:

1. Considering that the majority of the proofs of the stochastic approximation results are borrowed in part or in whole from Lakshminarayanan and Bhatnagar (2017), it seems that the methods of this paper are just an obvious extension of their methods, and therefore not of much interest. Can the authors clarify which parts of their methods they think are the most clever/interesting? Note: I would certainly consider it interesting if the authors can correctly address point #5 in my response to the previous question, as it would correct a major problem in Lakshminarayanan and Bhatnagar (2017).

2. In remark 2, the authors correctly point out that assumption 3 is very restrictive. Given that there are now works like Liu et al. (2025) and Mahadevan et al. (2026) which address the Markovian case in single- and two-timescale stochastic approximation, why should the multi-timescale case with i.i.d. noise be considered interesting? It seems that one should extend Mahadevan et al. (2026) to $N$-timescales instead of extending Lakshminarayanan and Bhatnagar (2017) to $N$-timescales, as this would enable us to avoid having to take assumption 3, and allow us to analyze the actual algorithms which use Markovian noise.

In my opinion, the most interesting part of this work is the inclusion of several applications and experiments. Previously, I didn’t think there would be any interesting algorithms that used 3 or 4 timescales.

**Broader Impact Concerns:**

None.

**Claims And Evidence:**

No

**Claims Explanation:**

This is a soft no. There are several technical problems, which range from relatively benign to very critical. Here are some issues with the stability proof:

1. The definition of $r(n)$ is inconsistent. Compare on page 9 in the paragraph labeled “Bounding the iterates: a recursive scaling argument” with D3 on page 28.

2. There is a discrepancy between the statement of Lemma 3.i and the first line of the proof. In the statement, the assumption is “if $||\bar{x}^{(1)}(T_n)|| > C_1(||\bar{x}^{(1)}(T_n)||\ldots$” while in the proof it is “if $||\bar{x}^{(1)}(T_n)|| > C_1(1+||\bar{x}^{(1)}(T_n)||\ldots$” I assume the authors intend the latter.

3. Stylistic-it might be cleaner if the authors could just use properties of norms (positive definite, subadditivity, absolute homogeneity) rather than treating these as $p$-norms and using properties of those. Both the proof of Lemma 6.i in Lakshminarayanan and Bhatnagar (2025) and the proof of Lemma 4.7 (found on page 22) of Mahadevan et al. (2026) prove essentially identical results to Lemma 3.i in this paper while maintaining the generality of arbitrary norms.

4. It seems wrong to just state “Here we focus on the case when iterates are blowing up” and then assume $r(n) = ||\bar{X}^{(1)}(T_n)||$. The sequence $r(n)$ (for contradiction) only needs to have a subsequence blowing up to infinity. So it seems that justification is required as to why Lemma 3.i holds $\textbf{for all}$ n. For instance, one could have for some n $r(n) = r(n+1) >> ||\bar{X}^{(1)}(T_{n+1})|| > ||\bar{X}^{(1)}(T_n)||$.

5. I think the above points are all somewhat easily fixable. However, there seems to be a $\textbf{critical problem}$ in the proof of Lemma 3.ii. The second to last sentence in the proof highlights that $||\bar{x}^{(1)}(T_n)||$ can’t grow by more than $K^{(1)}$ in one time period, which is true, but this does not entail a contradiction. Specifically, this does not immediately imply that the $\textbf{ratio}$ of $\frac{||\bar{x}^{(1)}(T_n)||}{(||\bar{x}^{(1)}(T_n)|| + \ldots + ||\bar{x}^{(N)}(T_n)||)}$ can’t grow by more than $K^{(1)}$ in one time period.
If $(||\bar{x}^{(1)}(T_n)|| + \ldots + ||\bar{x}^{(N)}(T_n)||)$ decreases, that also would increase the aforementioned ratio. The argument that $(||\bar{x}^{(1)}(T_n)|| + \ldots + ||\bar{x}^{(N)}(T_n)||)$ is quasi-static is also insufficient since it is in fact not static, and if it started close to 0, even a very small decrease in its size would greatly increase the ratio $\frac{||\bar{x}^{(1)}(T_n)||}{(||\bar{x}^{(1)}(T_n)|| + \ldots + ||\bar{x}^{(N)}(T_n)||)}$.
The correctness of this result is critical in this work, as without it, the stability argument doesn’t hold. I am aware that this argument is mostly borrowed from Lakshminarayanan and Bhatnagar (2017), but I think it is an error in their work as well.

These are the main issues I spotted, but I didn’t look at the proofs of convergence or the applications with as much scrutiny. However, the quantity of issues found in just one (crucial) part of the work does not engender much confidence in the correctness of the other sections.

**Requested Changes:**

Here are my requested changes:

1. Review the ICML paper Mahadevan et al. (2026) (https://arxiv.org/abs/2605.31172). It concerns stability and convergence of two-timescale stochastic approximation with Markovian noise, so it is a relevant related work, and some of its methods may be useful to the authors. It is a very recent work, so, understandably, it was not mentioned, but now that the authors are aware of it, they should include it.

2. Address the 5 technical concerns with the stability proof. Number 5 in particular is a  necessary fix.

3. Thoroughly proofread the paper and the proofs for other errors.

---

### Review · Reviewer_HEtu · 2026-07-05

**Summary Of Contributions:**

The paper studies stability and almost sure convergence of general N-timescale stochastic approximation recursions. The main theorem combines a convergence-under-stability result with a stability result based on scaled limiting ODEs. The paper then applies the framework to reinforcement-learning examples, including Gradient TD with heavy-ball momentum and constrained actor-critic.

The topic is relevant to stochastic approximation and reinforcement learning theory. A complete arbitrary fixed-N stability theorem could be useful as a reference result.

**Audience:**

Yes

**Audience Explanation:**

The potential value of the paper is mainly as a formalization and unification result: it organizes sufficient conditions for stability and almost sure convergence of unprojected martingale-noise multi-timescale SA recursions.

The paper addresses a topic that is relevant to a subset of the TMLR audience working on stochastic approximation, reinforcement learning theory, and multi-timescale learning algorithms. A unified fixed finite-N stability and convergence theorem could be useful as a reference point for researchers designing algorithms with more than two or three timescales.

**Broader Impact Concerns:**

I do not see significant broader impact concerns specific to this paper.

**Claims And Evidence:**

No

**Claims Explanation:**

My concern is whether the manuscript’s novelty and application-scope claims are supported with sufficient precision.

First, the manuscript repeatedly frames its contribution as providing the first sufficient conditions for stability and almost sure convergence of multi-timescale stochastic approximation recursions. This claim seems too broad without a clearer discussion of prior three-timescale work. In particular, Deb and Bhatnagar (AAAI 2022), Gradient Temporal Difference with Momentum: Stability and Convergence, already claims general three-timescale stability/convergence conditions and applies them to heavy-ball Gradient TD. Since the present manuscript also contains a closely related GTD momentum analysis and similar RL experiments, the paper should clearly identify which theoretical statements, algorithmic analysis, and experiments are new relative to that work.

Second, the paper advertises several reinforcement-learning applications, but one of them is not clearly supported by the theorem as stated. In the constrained actor-critic example, the multiplier recursion is projected onto $R_+^N$, while the main theorem is stated for ordinary additive SA recursions. Projection onto $R_+^N$ is standard in constrained stochastic approximation, but it does not by itself imply boundedness, and it may require a projected-ODE.

Thus, I do not think the current manuscript supports all of its novelty and application-scope claims with sufficiently accurate, convincing, and clear evidence.

**Requested Changes:**

## Major concern 1: Scope of the novelty claim

The manuscript clearly decomposes the analysis into convergence assuming stability and a separate stability argument. This decomposition is helpful. However, the current contribution statement seems broader than what the proof structure supports.

The convergence-under-stability part appears to be a finite-$N$ version of the standard multi-timescale ODE cascade argument. The proof itself states that it extends Borkar’s argument  (Borkar, 2008). I therefore view this part mainly as a completeness component rather than the main novelty.

The stability part is more central. My understanding is that it extends the two-timescale scaled-ODE stability argument of  Lakshminarayanan and Bhatnagar (2017) to arbitrary fixed finite $N$. This extension requires organization of the intermediate timescales, but the manuscript should make clearer what is new in the $N$-timescale formulation beyond this recursive extension.

Could the authors clarify the following?

- What is the precise technical increment of the stability proof over the two-timescale stability criterion of Lakshminarayanan and Bhatnagar (2017)?

- From my reading, Lemma 6 appears to be the main $N$-specific proof component, because it handles the induction over intermediate timescales. What is the main non-routine difficulty handled by this lemma?

## Major concern 2: Relation to prior three-timescale work

The stability theorem appears to be the core contribution of the manuscript. However, closely related prior work has already studied general three-timescale SA stability/convergence conditions.
In particular, the AAAI 2022 paper *Gradient Temporal Difference with Momentum: Stability and Convergence* already claims general three-timescale stability and convergence conditions, applies them to heavy-ball Gradient TD, and includes standard RL experiments (Deb and Bhatnagar, 2022). The present manuscript also analyzes Gradient TD with heavy-ball momentum and gives three-timescale GTD2-M/TDC-M decompositions. These parts appear closely related to that prior work.

Could the authors clarify the relationship between the present manuscript and the AAAI 2022 paper? In particular:

- Which theoretical statements in the current paper are new relative to the three-timescale stability/convergence conditions claimed in Deb and Bhatnagar (2022)? Since the apparent theoretical increment is from three timescales to arbitrary fixed finite $N$, what is technically non-routine in the $3 \to N$ step?

- Which parts of the GTD momentum analysis and experiments in Section 5.1 are new relative to that work?

## Major concern 3: Projected multiplier recursion in the constrained actor-critic example

The constrained actor-critic example in Section 5.3 uses a projected multiplier update. The projection onto $R_+^N$ is standard in constrained stochastic approximation, and it may be possible to handle it using standard projected-ODE arguments. However, the current main theorem is stated for ordinary additive SA recursions, while the multiplier update in Section 5.3 is projected. Since the projection is onto the noncompact set $R_+^N$, it does not by itself imply boundedness.

Could the authors clarify whether the projection is eventually inactive, whether a projected-ODE argument is being invoked, or whether this application is outside the direct scope of Theorem 1?


## References
- Borkar, V. S. (2008). *Stochastic Approximation: A Dynamical Systems Viewpoint*. Cambridge University Press.

- Lakshminarayanan, C. and Bhatnagar, S. (2017). A stability criterion for two timescale stochastic approximation schemes. *Automatica*.

- Deb, R. and Bhatnagar, S. (2022). Gradient Temporal Difference with Momentum: Stability and Convergence. *Proceedings of the AAAI Conference on Artificial Intelligence*.

---

### Review · Reviewer_sP7K · 2026-07-10

**Summary Of Contributions:**

The paper studies the stability and almost sure convergence of general \(N\)-timescale stochastic approximation. Convergence analyses for multi-timescale SA usually take boundedness of the iterates as given, and for \(N>2\), there has been no general way to deduce boundedness from checkable dynamical-systems conditions.

The paper uses the ODE method and splits the argument into two parts:
- (i) assuming boundedness, it eliminates the faster variables one layer at a time to obtain the nested equilibria and the final limit;
- (ii) it establishes boundedness of each layer through scaled trajectories and a recursive contraction argument.
The paper also covers asymptotically vanishing perturbations and applies the framework to GTD with momentum, off-policy actor–critic, and constrained actor–critic.

The main strength is that stability and conditional convergence are handled separately, which makes the proof structure clear. The recursive scaling argument is the most valuable technical piece, and th. three RL examples illustrate the kinds of algorithms the framework can cover. The main limitation of the current version is that several key steps in the general \(N\)-layer proof are hard to check, and the two actor–critic applications do not fully verify the conditions needed to invoke the main theorem.

**Additional Comments:**

The paper addresses a clear and useful problem. My current assessment is driven mainly by the completeness of the proofs and the scope of the applications discussed above.

Typo Lists:
- In the perturbed recursion in §3 (p. 7), the 2nd through $N$th updates are all written with $\varepsilon_n^{(1)}$; they should read $\varepsilon_n^{(2)},\ldots,\varepsilon_n^{(N)}$.
- (B.3.2) in §2.1 (p. 4) says "$j=3,4$", but there is no fourth layer in the three-timescale setting.
- On p. 26, in the analysis of the $\alpha_n^{(m)}$ timescale, the recursion is first written for $x_n^{(m)}$, then switches to $(x_n^{(m+1)},\ldots,x_n^{(N)})$ and $h^{(m+1)}$, and later switches back to $\lambda^{(m)}$ and $x_n^{(m)}$. Rewriting the slower updates as $\alpha_n^{(m)}(\alpha_n^{(j)}/\alpha_n^{(m)})$ is the usual change-of-clock trick, but the $m$ vs. $m+1$ indexing here should be made consistent.
- The order of the exponents in Assumption 7 of §5.2 (p. 17) is reversed. Since $\beta_t/\alpha_t\to0$ and $\eta_t/\beta_t\to0$, it should be $1/2<a<b<c\leq1$, not $1/2<c<b<a\leq1$.
- On pp. 7, 12, and 13, (A:2) and (A:3) are swapped in several places: the noise and its conditional second moment correspond to (A:2), while the step-size conditions correspond to (A:3).
- The start of §5 (p. 9) says the section analyzes "two RL algorithms", but §5.1–§5.3 in fact cover three.
- Near the end of the second-to-last paragraph of §6 (p. 23), "NN N-timescale stochastic approximation" should be "$N$-timescale stochastic approximation".
- The behavior/target policy notation and the direction of the importance ratio in §5.1 (p. 10) are inconsistent; they should follow the notation already used in §5.2 (p. 16): target $\pi$, behavior $b$, and $\rho_t=\pi(a_t\mid s_t)/b(a_t\mid s_t)$.

**Audience:**

Yes

**Audience Explanation:**

Multi-timescale SA is a common analysis tool in actor-critic and primal-dual RL, in optimization algorithms with auxiliary variables, and in hierarchical control. This should be of interest to readers at TMLR working on stochastic approximation and RL theory.

**Claims And Evidence:**

No

**Claims Explanation:**

The overall proof strategy is sound, and the linear GTD example does give a fairly complete verification of the conditions. However, the issues below currently prevent me from checking several of the main claims.

### 1. Checkability of the general $N$-layer stability proof
The key difficulty of the main theorem is the extension from two timescales to arbitrary $N$, yet the appendix lemmas that carry the scaled-ODE tracking—Lemmas 1-2, 4-5, and 7-8—are not proved in full: Lemmas 1-2 are on p. 29, Lemmas 4-5 on p. 32, and Lemmas 7-8 on p. 35, and in several places the proof is simply said to follow as in Lakshminarayanan & Bhatnagar (2017).

The steps said to follow "as in" the two-timescale case are not purely routine and should be spelled out, above all the intermediate-layer induction on pp. 32-34: layer $l$ is slower than the preceding $l-1$ layers and faster than the ones after it, and the two-timescale result does not directly cover this induction. In particular, the paper needs to explain how the ODE tracking in Lemma 5 combines with the induction hypothesis so that the recursive norm bound in Lemma 6,
$
\|x_n^{(l)}\|\leq K_l^*\sum_{j>l}\|x_n^{(j)}\|,
$
closes for every $\ell$.

### 2. Statement vs. verification for the off-policy actor-critic
The formal claim and the verification for the off-policy actor-critic are also inconsistent. On p. 7, §3.1 states that the actor converges almost surely to a stationary point with $\nabla_\theta J(\theta^*)=0$; on p. 19, §5.2 actually analyzes an ODE with an $\ell_2$ regularizer and only concludes that its trajectories approach a neighbourhood of the stationary set of the unregularized dynamics. The latter does not imply the former.

Moreover, the intermediate-timescale equilibrium on p. 19 relies on
$\bar A_z-\gamma\bar G_z\bar C^{-1}\bar A_z$
being nonsingular, but nonsingularity only lets you write down the equilibrium; it does not make the linear ODE globally asymptotically stable, which requires the coefficient matrix to be Hurwitz.

The paper also does not verify the slow-timescale condition (C.3.3) needed for Theorem 1, i.e., that the regularized actor ODE has a unique globally asymptotically stable equilibrium. So the paper needs to add the missing conditions and proofs here, or narrow the application claim on p. 7 to what is actually established.


### 3. Perturbation condition in Theorem 4
The perturbation condition in Theorem 4 (p. 7) is written as $\sum_{i=1}^N \varepsilon_n^{(i)} \to 0$. In general the $\varepsilon_n^{(i)}\in R^{d_i}$ may have different dimensions, so the sum need not even be well defined; and even when the dimensions agree, the condition allows non-vanishing perturbations in different coordinates to cancel.

For example, take two independent one-dimensional stable recursions with drifts $-x^{(1)}$ and $-x^{(2)}$, and set $\varepsilon_n^{(1)}=1,\varepsilon_n^{(2)}=-1$; the sum of the perturbations is identically zero, but the limit is no longer the equilibrium of the unperturbed system. The condition that makes sense here is $\|\varepsilon_n^{(i)}\|\to0$ for each $i$—equivalently, for fixed finite $N$, $\sum_i\|\varepsilon_n^{(i)}\|\to0$. Remark 1 on p. 9 only gestures at the extension with "similar arguments" and does not settle this. The issue most likely comes from notation or typesetting, but as written the theorem does not hold, so it should still be fixed.


### 4. Projection in the constrained actor-critic
The multiplier update of the constrained actor-critic contains a projection $\Gamma$ in Eq. (37) on p. 20 and Eq. (40) on p. 21. Theorem 1 analyzes an unprojected SA recursion in Euclidean space, whereas a projected update normally corresponds to a projected ODE or a differential inclusion. Even when the projection set is large, the projection cannot simply be ignored unless one can show it never becomes active along the subsequent trajectory; here the projection is onto $R_+^N$, whose boundary is directly tied to the KKT complementarity conditions. The authors need to show the projection is eventually inactive, or use an analysis that applies to projected recursions and re-establish the slow-timescale result accordingly.

### 5. The scaled Gibbs-policy limit near ties
Finally, §5.2 on p. 18 claims that the Gibbs policy $\pi_{cz}$ converges uniformly on compacts to the greedy policy, and uses this to verify (B.3.1); §5.3 on pp. 21-22 uses the same argument.

This uniform convergence generally fails near an action tie. In the simplest case, consider two actions with logits $z$ and 0: $\text{softmax}(cz)$ tends to pick the first action when $z>0$, the second when $z<0$, and stays at an even split when $z=0$, so the limit is discontinuous at $z=0$; a family of continuous functions cannot converge uniformly to such a discontinuous limit on any compact set containing 0. The authors should add a uniform action-gap assumption that rules out ties, use a weaker notion of convergence that still suffices for the argument, or rework the scaled limit.

For these reasons I currently choose No.

**Requested Changes:**

1. The omitted steps of the general $N$-layer stability proof are the main thing I would like to see, in particular the intermediate-layer induction in Lemmas 5-6 (pp. 32-34).

2. The off-policy actor-critic claim would need to match what is proved: either the coefficient matrix is shown Hurwitz and (C.3.3) is verified, or the stationary-point claim on p. 7 (and in the abstract/conclusion) is narrowed.

3. The perturbation condition in Theorem 4 (p. 7) looks incorrect as stated; $\|\varepsilon_n^{(i)}\|\to0$ for each $i$ would work, and the momentum GTD applications on pp. 12-14 would then need to be checked against it.

4. The projected multiplier recursion on pp. 20-22 still needs to be justified under the theory it invokes; otherwise the application and the KKT conclusion would have to be narrowed.

5. The scaled Gibbs-policy limit on p. 18 and pp. 21-22 does not seem to hold near action ties, so this part would need reworking.